# Distributionally Robust Cooperative Multi-Agent Reinforcement Learning via Robust Value Factorization

**Chengrui Qu**[*]
Caltech

**Christopher Yeh**[*]
Caltech

**Kishan Panaganti**[†]
Tencent AI Lab

**Eric Mazumdar**[*]
Caltech

**Adam Wierman**[*]
Caltech

## Abstract

Cooperative multi-agent reinforcement learning (MARL) commonly adopts centralized training with decentralized execution, where value-factorization methods enforce the individual-global-maximum (IGM) principle so that decentralized greedy actions recover the team-optimal joint action. However, the reliability of this recipe in real-world settings remains unreliable due to environmental uncertainties arising from the sim-to-real gap, model mismatch, and system noise. We address this gap by introducing **D**istributionally **r**obust **IGM** (DrIGM), a principle that requires each agent's robust greedy action to align with the robust team-optimal joint action. We show that DrIGM holds for a novel definition of robust individual action values, which is compatible with decentralized greedy execution and yields a provable robustness guarantee for the whole system. Building on this foundation, we derive DrIGM-compliant robust variants of existing value-factorization architectures (e.g., VDN/QMIX/QTRAN) that (i) train on robust Q-targets, (ii) preserve scalability, and (iii) integrate seamlessly with existing codebases without bespoke per-agent reward shaping. Empirically, on high-fidelity SustainGym simulators and a StarCraft game environment, our methods consistently improve out-of-distribution performance.

## 1 Introduction

Multi-agent reinforcement learning (MARL) is a popular framework for studying how multiple agents compete or cooperate in complex environments such as video game playing (Vinyals et al., 2019), economic policy design (Zheng et al., 2022), wireless network communications (Qu et al., 2020), and power grid control (Gao et al., 2021), among others. In this work, we focus on the *cooperative* MARL setting, where each agent can only observe its local history, and agents must collaborate to achieve a joint goal. To address partial observability and reduce real-time communication costs, a widely used paradigm is centralized training with decentralized execution (CTDE) (Oliehoek et al., 2008). During training, agents may aggregate global information, coordinate credit assignment, and learn a team structure; at deployment, each agent must act myopically based on its own local history.

The CTDE paradigm is typically realized through value factorization methods (e.g., VDN (Sunehag et al., 2017), QMIX (Rashid et al., 2020), QTRAN (Son et al., 2019)). A key concept that underpins the success of these methods is the individual-global-maximum (IGM) principle (Son et al., 2019), which aligns each agent's greedy action with the team-optimal joint action via a suitable value factorization. However, most examples where the success of this principle is demonstrated are in virtual tasks (e.g., games (Vinyals et al., 2017) and grid worlds (Leibo et al., 2017)). It remains unclear whether this principle maintains its reliability in real-world domains, where modeling is imperfect and execution is noisy.

In practice, a major obstacle facing cooperative MARL is environmental uncertainty (Shi et al., 2024): team performance can drop sharply when the deployed environment deviates from the training

---

[*]Department of Computing Mathematical Sciences, California Institute of Technology, CA 91125, USA
[†]Tencent AI Lab in Bellevue, WA, USA

environment due to model mismatch, system noise, and the sim-to-real gap (Zhang et al., 2020b; Balaji et al., 2019). While environmental uncertainty presents challenges in single-agent RL settings, it is a more significant hurdle in cooperative MARL, where partial observability and inter-agent coupling can cause small mismatches to cascade into coordination failures (Capitan et al., 2012; He et al., 2022).

In single-agent RL, uncertainty in the environment is commonly addressed by distributionally robust RL (DR-RL) techniques (Wiesemann et al., 2013; Taori et al., 2020; Nilim & El Ghaoui, 2005; Panaganti & Kalathil, 2021a; Shi et al., 2023) which seek policies that perform well under adversarial perturbations of a nominal environment model. Single-agent DR-RL is well-explored, however extending DR-RL to the cooperative MARL setting is fundamentally more challenging. In particular, each agent acts on a local history yet shares a team reward coupled with teammates' actions, making it nontrivial to define individual robust Q-functions that both evaluate worst-case outcomes and remain compatible with IGM for decentralized greedy execution. Reward engineering can help empirically, but only a narrow class of shaping functions can provably preserve optimality (Foerster et al., 2016), even in the single-agent setting. Thus, *we seek a principled route to distributional robustness for cooperative MARL that remains compatible with decentralized greedy execution.*

**Contributions.** In this paper, we introduce a family of distributionally robust cooperative MARL algorithms for the CTDE setting. Our central technique is **D**istributionally **r**obust **IGM** (DrIGM), a robustness principle that requires each agent's robust greedy action to coincide with the robust team-optimal joint action, thereby preserving decentralized greedy execution.

We first show, via a concrete counterexample, that naïvely adopting individual robust action-value functions from single-agent DR-RL, where each agent considers its own worst case, does *not* guarantee decentralized alignment (IGM) in the cooperative multi-agent setting. We then provide a sufficient condition under which DrIGM holds: when individual robust action-value functions are defined with respect to the worst-case joint action-value function, DrIGM is guaranteed.

Next, we derive DrIGM-compliant robust variants of existing value factorization architectures (VDN, QMIX, and QTRAN), by training on robust Q-targets while retaining the CTDE information structure. The resulting methods are scalable, easy to implement on top of existing codebases, and maintain robustness at execution without requiring bespoke individual robust value design.

Finally, we evaluate our DrIGM-based algorithms on a realistic simulation an HVAC control task in SustainGym (Yeh et al., 2023), as well as on SMAC, a StarCraft II-based multi-agent game-playing environment (Samvelyan et al., 2019). Across out-of-distribution settings, our methods outperform non-robust value factorization baselines and a recent robust cooperative MARL baseline, consistently mitigating sim-to-real degradation on operational metrics.

**Brief discussion of related work.** Robustness in cooperative MARL has been studied along several axes: adversarial or heterogeneous teammates (Li et al., 2019; Kannan et al., 2023; Li et al., 2024), state/observation and communication perturbations (Guo et al., 2024; Yu et al., 2024), risk-sensitive (tail-aware) objectives under a fixed model (Shen et al., 2023), and explicit model uncertainty (Kwak et al., 2010; Zhang et al., 2020b; 2021; Liu et al., 2025). Most of the works on model uncertainty adopt a distributionally robust optimization viewpoint and targets Nash solutions with provable algorithms, often assuming full observability or individual rewards (Zhang et al., 2020a; Kardeş et al., 2011; Ma et al., 2023; Blanchet et al., 2023; Shi et al., 2024; Liu et al., 2025). In this work, we focus on the cooperative CTDE regime with partial observability and a single team reward, providing a systematic framework for robustness to model uncertainty without real-time communication. Due to space constraints, we provide an extended discussion of related works in Appendix A.

## 2 BACKGROUND AND PROBLEM FORMULATION

**Notation.** For a set $\mathcal{X}$, $|\mathcal{X}|$ denotes its *cardinality* and $\Delta(\mathcal{X})$ the probability simplex over $\mathcal{X}$. We write $\prod_i \mathcal{X}_i$ for the Cartesian product. For $N \in \mathbb{N}$, we let $[N] := \{1, \dots, N\}$. Let $[x]_+ := \max\{0, x\}$.

**Cooperative Dec-POMDPs.** A cooperative multi-agent task with $N$ agents is modeled as a Decentralized Partially Observable Markov Decision Process (Dec-POMDP):

$$G = (\mathcal{S}, \{\mathcal{A}_i\}_{i=1}^N, P, r, \{\mathcal{O}_i\}_{i=1}^N, \{\sigma_i\}_{i=1}^N, \gamma),$$

with joint action space $\mathcal{A} := \prod_{i \in [N]} \mathcal{A}_i$. At time $t$, each agent $i$ obtains an individual observation $o_i^t := \sigma_i(s^t)$ from its observation space $\mathcal{O}_i$, chooses an action $a_i^t \in \mathcal{A}_i$, a bounded joint reward $r(s^t, \mathbf{a}^t)$ is received, where $\mathbf{a}^t := (a_1^t, \ldots, a_N^t) \in \mathcal{A}$ is the joint action, and then the state evolves via $s^{t+1} \sim P(\cdot \mid s^t, \mathbf{a}^t)$. Here we assume the joint observation $(o_1, \ldots, o_N)$ can recover the full state.[1] Each agent $i$ acts using a history-based policy $\pi_i(\cdot \mid h_i^t)$ with $h_i^t := (o_i^0, a_i^0, \ldots, o_i^{t-1}, a_i^{t-1}, o_i^t)$; the joint policy is $\pi = \langle \pi_1, \ldots, \pi_N \rangle$. We use $\mathcal{H}_i^t$ to denote the space of possible histories for agent $i$ up to time $t$. In the following sections, we will omit the superscript $t$ to avoid notational clutter. The joint action-observation history is denoted $\mathbf{h} \in \mathcal{H} := \prod_{i \in [N]} \mathcal{H}_i$. Given the current state $s$, the joint history $\mathbf{h}$ and the joint action $\mathbf{a}$, we denote the joint action-value function under policy $\pi$ by $Q_{\text{tot}}^{P,\pi}(\mathbf{h}, \mathbf{a})$, which can be reduced to $|\mathcal{S} \times \mathcal{A}|$ dimension as we assume the joint observation can recover the full state. We use $Q_{\text{tot}}^P(\mathbf{h}, \mathbf{a}) := \max_\pi Q_{\text{tot}}^{P,\pi}(\mathbf{h}, \mathbf{a})$ to denote the optimal joint action value.

**CTDE.** Centralized training with decentralized execution (CTDE) leverages global information during learning while executing individual policies from individual histories. A common CTDE approach is value factorization, which learns an optimal joint action-value $Q_{\text{tot}}^P(\mathbf{h}, \mathbf{a})$ and individual action-value functions $[Q_i^P(h_i, a_i)]_{i \in [N]}$ that satisfy the following *individual-global-max* (IGM) principle (Son et al., 2019).

**Definition 1** (IGM). *We say that individual action-value functions $[Q_i^P : \mathcal{H}_i \times \mathcal{A}_i \to \mathbb{R}]_{i \in [N]}$ satisfy the* individual-global-max (IGM) principle *for an optimal joint action-value function $Q_{\text{tot}}^P : \mathcal{H} \times \mathcal{A} \to \mathbb{R}$ under joint history $\mathbf{h} = (h_1, \ldots, h_N) \in \mathcal{H}$ if*

$$\left( \underset{a_1}{\arg\max} \, Q_1^P(h_1, a_1), \ldots, \underset{a_N}{\arg\max} \, Q_N^P(h_N, a_N) \right) \subseteq \underset{\mathbf{a}}{\arg\max} \, Q_{\text{tot}}^P(\mathbf{h}, \mathbf{a}).$$

IGM ensures that greedy individual actions are jointly optimal w.r.t. $Q_{\text{tot}}^P$, enabling decentralized execution without test-time communication.

**Robust Dec-POMDPs.** To capture model error and deployment shift, we consider an *uncertainty set $\mathcal{P}$* of environment models around a nominal $P^0$. In Dec-POMDPs, we work with a history-based view: let $P_{\mathbf{h},\mathbf{a}}(\cdot)$ denote the transition kernel over next joint histories $\mathbf{h}'$, given the current joint history $\mathbf{h}$ and the joint action $\mathbf{a}$. We assume a *history-action rectangular* uncertainty set.

$$\mathcal{P} = \prod_{(\mathbf{h},\mathbf{a}) \in \mathcal{H} \times \mathcal{A}} \mathcal{P}_{\mathbf{h},\mathbf{a}}, \quad \mathcal{P}_{\mathbf{h},\mathbf{a}} \subseteq \Delta(\mathcal{H}), \tag{1}$$

e.g., balls around $P_{\mathbf{h},\mathbf{a}}^0$ under a probability metric with radius $\rho > 0$. This rectangularity assumption is widely adopted in DR-RL literature (Blanchet et al., 2023; Shi et al., 2024; Ma et al., 2023), and ensures that a robust policy exists.

Given a function $Q : \mathcal{H} \times \mathcal{A} \to \mathbb{R}$, define the *robust Bellman operator $\mathcal{T}$* as

$$(\mathcal{T}Q)(\mathbf{h}, \mathbf{a}) := r(s, \mathbf{a}) + \gamma \inf_{P_{\mathbf{h},\mathbf{a}} \in \mathcal{P}_{\mathbf{h},\mathbf{a}}} \mathbb{E}_{\mathbf{h}' \sim P_{\mathbf{h},\mathbf{a}}} \left[ \max_{\mathbf{a}' \in \mathcal{A}} Q(\mathbf{h}', \mathbf{a}') \right]. \tag{2}$$

Under standard assumptions (bounded rewards, $\gamma \in (0,1)$) and rectangularity in Eq. (1), $\mathcal{T}$ is a $\gamma$-contraction on the space of bounded $Q$, so it has a unique fixed point $Q_{\text{tot}}^{\mathcal{P}}$ (Iyengar, 2005) which satisfies

$$Q_{\text{tot}}^{\mathcal{P}}(\mathbf{h}, \mathbf{a}) = \inf_{P \in \mathcal{P}} Q_{\text{tot}}^P(\mathbf{h}, \mathbf{a}), \qquad \forall (\mathbf{h}, \mathbf{a}) \in \mathcal{H} \times \mathcal{A}. \tag{3}$$

We call $Q_{\text{tot}}^{\mathcal{P}}$ the *optimal robust joint action-value function* for the Dec-POMDP, and it admits a deterministic robust greedy joint policy $\pi^*(\mathbf{h}) \in \arg\max_{\mathbf{a}} Q_{\text{tot}}^{\mathcal{P}}(\mathbf{h}, \mathbf{a})$.

**Robust Cooperative MARL.** We study robust cooperative MARL under the CTDE setting. Given a model uncertainty set $\mathcal{P}$, our goal is to learn decentralized policies that maximize $Q_{\text{tot}}^{\mathcal{P}}$. That is, we aim to find $[\pi_i^\star : \mathcal{H}_i \mapsto \mathcal{A}_i]_{i \in [N]}$, such that

$$\langle \pi_1^\star, \ldots, \pi_N^\star \rangle \in \underset{\mathbf{a}}{\arg\max} \, Q_{\text{tot}}^{\mathcal{P}}(\cdot, \mathbf{a}).$$

---

[1]Formally, we assume that $\sigma = (\sigma_1(\cdot), \ldots, \sigma_N(\cdot)) : \mathcal{S} \to \prod_{i \in [N]} \mathcal{O}_i$ is injective. Equivalently, there exists a (deterministic) decoding map $g : \prod_{i \in [N]} \mathcal{O}_i \to \mathcal{S}$ such that $g(\sigma(s)) = s$ for all $s \in \mathcal{S}$.

Specifically, we seek a value factorization method that automatically generates robust individual action values, thereby enabling a decentralized policy. This is non-trivial for two reasons. First, no individual reward signals are available, so robust individual action values are ill-defined a priori. Second, directly defining robust individual action values from the single-agent DR-RL literature can break standard value factorization. As we demonstrate in Example 1, robust individual actions may not align with the robust joint action. These challenges motivate the central question of our work: *Can we construct robust individual utilities and a mixing scheme such that decentralized greedy actions recover the joint maximizers of $Q_{\text{tot}}^{\mathcal{P}}$, thereby enabling a robust CTDE framework?*

## 3 DISTRIBUTIONALLY ROBUST IGM (DRIGM)

To address the question above, we propose a novel principle for robust value factorization that builds upon the IGM principle while explicitly incorporating robustness.

### 3.1 DISTRIBUTIONALLY ROBUST IGM (DRIGM) PRINCIPLE

**Definition 2** (DrIGM). *Given an uncertainty set $\mathcal{P}$, we say that **robust** individual action-value functions $[Q_i^{\text{rob}} : \mathcal{H}_i \times \mathcal{A}_i \to \mathbb{R}]_{i \in [N]}$ satisfy the **D**istributionally **r**obust **IGM** (DrIGM) principle for the optimal **robust** joint action-value function $Q_{\text{tot}}^{\mathcal{P}} : \mathcal{H} \times \mathcal{A} \to \mathbb{R}$ under joint history $\mathbf{h} = (h_1, \ldots, h_N) \in \mathcal{H}$ if*

$$\left( \arg\max_{a_1} Q_1^{\text{rob}}(h_1, a_1), \ldots, \arg\max_{a_N} Q_N^{\text{rob}}(h_N, a_N) \right) \subseteq \arg\max_{\mathbf{a}} Q_{\text{tot}}^{\mathcal{P}}(\mathbf{h}, \mathbf{a}).$$

DrIGM extends classical IGM to the robust setting by requiring that the *robust* joint greedy action induced by $Q_{\text{tot}}^{\mathcal{P}}$ factorizes into robust individual greedy actions from $[Q_i^{\text{rob}}]_{i \in [N]}$. Note that when $\mathcal{P} = \{P\}$ is a singleton (i.e., there is no uncertainty), then DrIGM is equivalent to IGM.

Satisfying DrIGM is nontrivial. In particular, the single-agent definition $Q_i^{\text{rob}}(s, a) = \inf_{P \in \mathcal{P}} Q_i^P(s, a)$, commonly adopted in the DR-RL literature, does not ensure DrIGM when applied with a global uncertainty set. As shown in Example 1 in Appendix B, an adversarial model $P \in \mathcal{P}$ that minimizes one agent's value need not coincide with the adversarial model $P' \in \mathcal{P}$ that minimizes the joint value. As a result, robust individual greedy actions may fail to align with the robust joint greedy action. Similar inconsistencies arise even under agent-wise uncertainty sets defined in Shi et al. (2024) for essentially the same reason. This highlights the need for a new formulation of robust individual action values to support a consistent robust CTDE framework.

In robust cooperative MARL, the primary concern is the robustness of the *entire system*, as opposed to robustness of individual agents. Thus, it is sufficient to consider the worst case for the joint action value, rather than independently for each agent. Motivated by this idea, we show that the robust individual action value defined under a global worst-case model can guarantee DrIGM.

**Theorem 1.** *Given a global uncertainty set $\mathcal{P}$ defined in Eq. (1), suppose for all $P \in \mathcal{P}$, there exist $[Q_i^P]_{i \in [N]}$ satisfying IGM for $Q_{\text{tot}}^P$ under joint history $\mathbf{h} = (h_1, \ldots, h_N) \in \mathcal{H}$. Let*

$$P^{\text{worst}}(\mathbf{h}, \mathbf{a}) \in \arg\inf_{P \in \mathcal{P}} Q_{\text{tot}}^P(\mathbf{h}, \mathbf{a}), \tag{4}$$

$$\bar{\mathbf{a}} \in \arg\max_{\mathbf{a}} Q_{\text{tot}}^{\mathcal{P}}(\mathbf{h}, \mathbf{a}), \tag{5}$$

*denote the global worst-case model and the robust joint greedy action, respectively. For each agent $i \in [N]$, define the robust individual action-value functions $Q_i^{\text{rob}}$ as*

$$Q_i^{\text{rob}}(h_i, a_i) := Q_i^{P^{\text{worst}}(\mathbf{h}, \bar{\mathbf{a}})}(h_i, a_i). \tag{6}$$

*Then, $[Q_i^{\text{rob}}]_{i \in [N]}$ satisfy DrIGM for $Q_{\text{tot}}^{\mathcal{P}}$ under joint history $\mathbf{h}$.*

The proof of Theorem 1 can be found in Appendix C.1. Theorem 1 demonstrates that by anchoring individual robust action values to the global worst-case model evaluated at the robust joint greedy action, individual robust greedy actions become aligned with the robust joint greedy action. This construction resolves the misalignment problem that occurs when individual adversaries differ from the global adversary. More broadly, the result highlights that robust CTDE is achieved not by independently robustifying each agent, but by coordinating all agents against a shared adversarial model tied to the team's worst-case joint outcome. This perspective offers a principled foundation for designing robust value factorization methods that maintain decentralized execution while ensuring robustness guarantees.

**Common factorization methods satisfy DrIGM.** Having established that robust individual action values can be consistently defined via Theorem 1, we now examine whether these values are compatible with standard factorization methods used in cooperative MARL.

**Theorem 2.** *Given $\mathcal{P}$ defined in Eq. (1), for a joint history $\mathbf{h} \in \mathcal{H}$, suppose for all $P \in \mathcal{P}$, there exist individual action-value functions $[Q_i^P]_{i \in [N]}$ satisfying one of the following conditions for all $\mathbf{a} = (a_1, \ldots, a_N) \in \mathcal{A}$:*

$$Q_{\text{tot}}^P(\mathbf{h}, \mathbf{a}) = \sum_{i \in [N]} Q_i^P(h_i, a_i), \tag{VDN}$$

$$\frac{\partial Q_{\text{tot}}^P(\mathbf{h}, \mathbf{a})}{\partial Q_i^P(h_i, a_i)} \geq 0, \quad \forall i \in [N], \tag{QMIX}$$

$$\sum_{i \in [N]} Q_i^P(h_i, a_i) - Q_{\text{tot}}^P(\mathbf{h}, \mathbf{a}) + V_{\text{tot}}(\mathbf{h}) = \begin{cases} 0, & \mathbf{a} = \bar{\mathbf{a}}, \\ \geq 0, & \mathbf{a} \neq \bar{\mathbf{a}}, \end{cases} \tag{QTRAN}$$

*where $\bar{\mathbf{a}} := [\bar{a}_i]_{i \in [N]}$ with $\bar{a}_i := \arg\max_{a_i} Q_i^P(h_i, a_i)$ and $V_{\text{tot}}(\mathbf{h}) := \max_{\mathbf{a}} Q_{\text{tot}}^P(\mathbf{h}, \mathbf{a}) - \sum_{i \in [N]} Q_i^P(h_i, a_i)$ with $\mathbf{a} = [a_i]_{i \in [N]}$. Then $[Q_i^{\text{rob}}]_{i \in [N]}$ as defined in Eq. (6) satisfy DrIGM for $Q_{\text{tot}}^{\mathcal{P}}$ under joint history $\mathbf{h}$.*

The proof of Theorem 2 can be found in Appendix C.2. Theorem 2 shows that when the underlying individual Q-functions satisfy the structural conditions of VDN (Sunehag et al., 2017), QMIX (Rashid et al., 2020), or QTRAN (Son et al., 2019), the robust individual action values $[Q_i^{\text{rob}}]_{i \in [N]}$ constructed from Eq. (6) automatically satisfy DrIGM. This result ensures that robust CTDE can be realized directly within widely used value factorization frameworks, enabling principled distributionally robust extensions of existing algorithms. Moreover, as long as the test environment lies within the prescribed uncertainty set, this approach yields a provable robustness guarantee, as formalized in the next theorem.

**Theorem 3.** *Given $\mathcal{P}$ defined in Eq. (1), suppose the robust individual action values $Q_i^{\text{rob}}$ satisfy Definition 2. If the test environment model $P_{\text{test}}$ is included in the uncertainty set (i.e., $P_{\text{test}} \in \mathcal{P}$), then the robust joint action values provably lower bound the real joint action values in $P_{\text{test}}$:*

$$Q_{\text{tot}}^{\mathcal{P}}(\mathbf{h}, \mathbf{a}) \leq Q_{\text{tot}}^{P_{\text{test}}}(\mathbf{h}, \mathbf{a}), \quad \forall \mathbf{h} \in \mathcal{H}, \ \mathbf{a} \in \mathcal{A}.$$

The proof of Theorem 3 can be found in Appendix C.3.

### 3.2 ROBUST BELLMAN OPERATORS UNDER SPECIFIC UNCERTAINTY SETS

To design training loss functions, we next present the DrIGM-based robust Bellman operators for two common uncertainty designs: $\rho$-contamination and total variation (TV), which are well-studied in the single-agent distributionally robust RL literature (Yang et al., 2022; Panaganti & Kalathil, 2021b; Xu et al., 2023; Dong et al., 2022; Liu & Xu, 2024; Panaganti et al., 2022; Wang & Zou, 2022; Zhang et al., 2024). Both types of uncertainty sets consider perturbations of size $\rho \in (0, 1]$ around a nominal model $P^0$.

The $\rho$-contamination uncertainty set is defined as (for all $\mathbf{h} \in \mathcal{H}$ and $\mathbf{a} \in \mathcal{A}$)

$$\mathcal{P}_{\mathbf{h}, \mathbf{a}} = \left\{ P \in \Delta(\mathcal{H}) \,\middle|\, P_{\mathbf{h}, \mathbf{a}} = (1 - \rho) P_{\mathbf{h}, \mathbf{a}}^0 + \rho \nu_{\mathbf{h}, \mathbf{a}}, \ \nu \in \Delta(\mathcal{H}) \text{ is arbitrary} \right\}, \tag{7}$$

with corresponding robust Bellman operator

$$(\mathcal{T} Q_{\text{tot}}^{\mathcal{P}})(\mathbf{h}, \mathbf{a}) \overset{(a)}{=} r(s, \mathbf{a}) + \gamma(1 - \rho) \mathbb{E}_{\mathbf{h}' \sim P_{\mathbf{h}, \mathbf{a}}^0} \left[ \max_{\mathbf{a}' \in \mathcal{A}} Q_{\text{tot}}^{\mathcal{P}}(h_1', \ldots, h_N', \mathbf{a}') \right]$$

$$\overset{(b)}{=} r(s, \mathbf{a}) + \gamma(1 - \rho) \mathbb{E}_{\mathbf{h}' \sim P_{\mathbf{h}, \mathbf{a}}^0} \left[ Q_{\text{tot}}^{\mathcal{P}}(h_1', \ldots, h_N', \bar{a}_1', \ldots, \bar{a}_N') \right], \tag{8}$$

where $\bar{a}_i' = \arg\max_{a_i'} Q_i^{\text{rob}}(h_i', a_i')$. Here, (a) follows from robust Bellman operator as in the single-agent setting due to the $\mathbf{h} \times \mathbf{a}$-rectangularity from Eq. (1). (b) follows from the DrIGM principle where robust individual greedy actions are aligned with the robust joint greedy action.

Similarly, the TV-uncertainty set is defined as (for all $\mathbf{h} \in \mathcal{H}$ and $\mathbf{a} \in \mathcal{A}$)

$$\mathcal{P}_{\mathbf{h}, \mathbf{a}} = \left\{ P \in \Delta(\mathcal{H}) \,\middle|\, \text{TV}(P, P_{\mathbf{h}, \mathbf{a}}^0) \leq \rho \right\}, \tag{9}$$

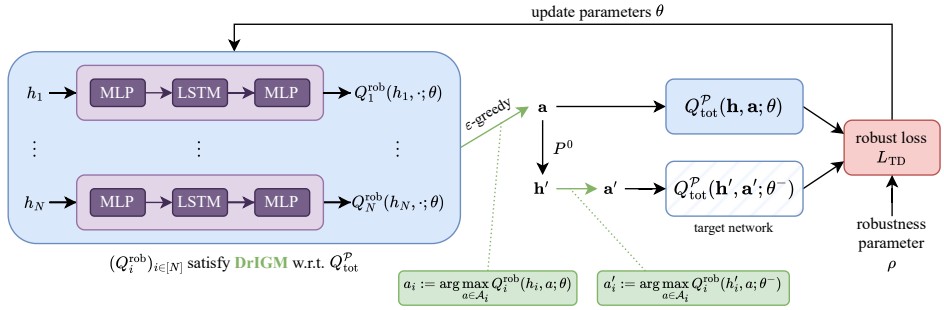

Figure 1: Overview of our robust value factorization algorithms. Because the robust individual action-value functions satisfy DrIGM, greedy actions can be computed efficiently in a decentralized manner while the function parameters are trained with a robust TD loss based on global reward.

with corresponding robust Bellman operator,

$$
(\mathcal{T}Q_{\text{tot}}^{\mathcal{P}})(\mathbf{h}, \mathbf{a}) = r(s, \mathbf{a}) - \inf_{\eta \in [0, \frac{2}{\rho(1-\gamma)}]} \gamma \mathbb{E}_{\mathbf{h}' \sim P_{\mathbf{h}, \mathbf{a}}^0} \Bigg( -(1-\rho)\eta(s, \mathbf{a})
$$
$$
+ \left[ \eta(s, \mathbf{a}) - Q_{\text{tot}}^{\mathcal{P}}(h_1', \dots, h_N', \bar{a}_1', \dots, \bar{a}_N') \right]_+ \Bigg), \tag{10}
$$

where $\eta(s, a)$ is the dual variable.

The derivation of the robust Bellman operators relies on a "fail state" assumption (Assumption 1); the proof is provided in Appendix D. In the next section, we show how DrIGM leads to practical robust value factorization algorithms.

## 4 ALGORITHMS: ROBUST VALUE FACTORIZATION

**Overall framework.** Guided by DrIGM, we develop six robust value factorization algorithms by combining two types of uncertainty sets ($\rho$-contamination, TV-uncertainty) with three different value factorization architectures (VDN, QMIX, QTRAN). The overall framework is illustrated in Algorithm 1 and Fig. 1, while detailed pseudocode for each variant can be found in Appendix E. Concretely, we collect trajectories using $\varepsilon$-greedy exploration and train the robust individual action-value network using TD-learning (Sutton & Barto, 2018). For stability, robust one-step targets are evaluated using *target* networks, which are updated periodically.

**Robust individual action-value networks.** Each agent $i$ uses a DRQN (Deep Recurrent Q Network)-style network that maps its local history $h_i$ (observations and past actions) to action-values $Q_i(h_i, a_i)$, following an *MLP encoder → LSTM core → MLP* output architecture. For our training procedure, we follow the approach from Hausknecht & Stone (2015). We sample mini-batches of *sub-trajectories* from the replay buffer $\mathcal{D}$ and use *bootstrapped random updates*. We use 8 burn-in steps to warm-start the LSTM state and only take the last step output to calculate the loss and update the networks. This procedure is computationally and memory efficient while achieving performance comparable to sequential updates from the start of each episode.

**Factorization networks.** We instantiate three networks for robust value factorization:

1. **VDN** factorizes the robust joint action-value as the sum of robust per-agent values,

$$
Q_{\text{tot}}^{\mathcal{P}, \text{VDN}}(\mathbf{h}, \mathbf{a}) = \sum_{i=1}^{N} Q_i^{\text{rob}}(h_i, a_i).
$$

2. Beyond direct summation, **QMIX** uses a *monotone* mixing network

$$
Q_{\text{tot}}^{\mathcal{P}, \text{QMIX}}(\mathbf{h}, \mathbf{a}) = f_\theta\big(Q_1^{\text{rob}}(h_1, a_1), \dots, Q_N^{\text{rob}}(h_N, a_N), s\big), \tag{11}
$$

where $s$ is the global state. A lightweight *hypernetwork* takes $s$ as input and outputs the layer weights of $f_\theta$; to ensure $\partial Q_{\text{tot}}^{\mathcal{P}, \text{QMIX}} / \partial Q_i^{\text{rob}} \geq 0$ (the QMIX monotonicity constraint),

---

**Algorithm 1** Robust value factorization

---

1: Input robustness parameter $\rho$, target network update frequency $f$, and $\varepsilon$
2: Initialize replay buffer $\mathcal{D}$
3: Initialize **robust individual action-value networks** $[Q_i^{\mathrm{rob}}]_{i \in [N]}$ with random parameters $\theta$
4: Initialize **factorization networks** that produce $Q_{\mathrm{tot}}^{\mathcal{P}}$ with random parameters $\theta$
5: Initialize target parameters $\theta^- = \theta$
6: **for** episode $e = 1, \ldots, E$ **do**
7:  Observe initial state $s^0$ and observation $o_i^0 = \sigma_i(s^0)$ for each agent $i$.
8:  **for** $t = 1, \ldots, T$ **do**
9:    Each agent $i$ chooses its action $a_i^t$ using $\varepsilon$-greedy policy.
10:    Take joint action $\mathbf{a}^t$, observe the next state $s^{t+1}$, reward $r^t$ and observation $o_i^{t+1} = \sigma_i(s^{t+1})$
     for each agent $i$
11:    Store transition $(\mathbf{h}^t, \mathbf{a}^t, r^t, \mathbf{h}^{t+1})$ in replay buffer $\mathcal{D}$
12:    Sample a mini-batch of transitions $(\mathbf{h}, \mathbf{a}, r, \mathbf{h}')$ from $\mathcal{D}$
13:    Calculate **TD loss** $L_{\mathrm{TD}}$ using Eq. (14)
14:    Update $\theta$ by minimizing $L_{\mathrm{TD}}$
15:    Update $\theta^- = \theta$ with frequency $f$
16:  **end for**
17: **end for**

---

we enforce elementwise nonnegativity on these weights via an absolute-value (or softplus) transform. Biases remain unconstrained.

3. **QTRAN** learns a separate joint action-value function $Q_{\mathrm{tot}}^{\mathcal{P},\mathrm{QTRAN}}(\mathbf{h}, \mathbf{a})$ and a baseline $V_{\mathrm{tot}}(\mathbf{h})$. For efficiency and scalability, the joint network shares the encoder/head with the individual DRQN modules. In addition to the robust TD loss, QTRAN imposes two *consistency* terms to align the factorized and joint values:

$$L_{\mathrm{opt}} = \left( Q_{\mathrm{tot}}^{\mathcal{P},\mathrm{VDN}}(\mathbf{h}, \bar{\mathbf{a}}) - \widehat{Q}_{\mathrm{tot}}^{\mathcal{P},\mathrm{QTRAN}}(\mathbf{h}, \bar{\mathbf{a}}) + V_{\mathrm{tot}}(\mathbf{h}) \right)^2, \tag{12}$$

$$L_{\mathrm{nopt}} = \left( \min\left[ Q_{\mathrm{tot}}^{\mathcal{P},\mathrm{VDN}}(\mathbf{h}, \mathbf{a}) - \widehat{Q}_{\mathrm{tot}}^{\mathcal{P},\mathrm{QTRAN}}(\mathbf{h}, \mathbf{a}) + V_{\mathrm{tot}}(\mathbf{h}), \, 0 \right] \right)^2, \tag{13}$$

where the $\hat{Q}$ is the detached Q value for training stability. Intuitively, $L_{\mathrm{opt}}$ enforces equality at the (robust) greedy joint action, while $L_{\mathrm{nopt}}$ penalizes positive slack elsewhere, recovering the QTRAN constraints in our robust setting.

**TD Loss.** Given the robust Bellman operator $\mathcal{T}$ defined in Eq. (2) for a Dec-POMDP setting, the generic form of the TD-loss is

$$L_{\mathrm{TD}} = \left( Q_{\mathrm{tot}}^{\mathcal{P}}(\mathbf{h}, \mathbf{a}; \theta) - (\mathcal{T} Q_{\mathrm{tot}}^{\mathcal{P}}(\cdot, \cdot; \theta^-))(\mathbf{h}, \mathbf{a}) \right)^2, \tag{14}$$

where $\theta$ is the network parameters, and $\theta^-$ is the target network parameters for training stability. Specifically, for $\rho$-contamination uncertainty sets, by the robust Bellman operator in Eq. (8), we have

$$L_{\mathrm{TD}} = \left( Q_{\mathrm{tot}}^{\mathcal{P}}(\mathbf{h}, \mathbf{a}; \theta) - (r(s, \mathbf{a}) + \gamma(1 - \rho)\, \mathbb{E}_{\mathbf{h}' \sim P_{\mathbf{h}, \mathbf{a}}^0}\, Q_{\mathrm{tot}}^{\mathcal{P}}(\mathbf{h}', \bar{\mathbf{a}}'; \theta^-)) \right)^2, \tag{15}$$

For TV uncertainty sets, by the robust Bellman operator in Eq. (10), we have

$$L_{\mathrm{TD}} = \left( Q_{\mathrm{tot}}^{\mathcal{P}}(\mathbf{h}, \mathbf{a}; \theta) - r(s, \mathbf{a}) + \gamma\, \mathbb{E}_{\mathbf{h}' \sim P_{\mathbf{h}, \mathbf{a}}^0} \left[ -(1 - \rho)\eta(s, \mathbf{a}) \right. \right.$$
$$\left. \left. + [\eta(s, \mathbf{a}) - Q_{\mathrm{tot}}^{\mathcal{P}}(\mathbf{h}', \bar{\mathbf{a}}'; \theta^-)]_+ \right] \right)^2, \tag{16}$$

where $\eta : \mathcal{S} \times \mathcal{A} \to \mathbb{R}$ is calculated by minimizing the following empirical loss:

$$L_{\mathrm{dual}}(\eta, Q_{\mathrm{tot}}^{\mathcal{P}}) = \frac{1}{|\mathcal{D}|} \sum_{(\mathbf{h}, \mathbf{a}, \mathbf{h}') \in \mathcal{D}} \left( \left[ \eta(s, \mathbf{a}) - \max_{\mathbf{a}'} Q_{\mathrm{tot}}^{\mathcal{P}}(\mathbf{h}', \mathbf{a}') \right]_+ - (1 - \rho)\, \eta(s, \mathbf{a}) \right). \tag{17}$$

## 5 EXPERIMENTS

### 5.1 SUSTAINGYM

We evaluate our proposed robust value factorization methods on SustainGym (Yeh et al., 2023), a recent benchmark suite designed to simulate real-world control tasks under distribution shift. We focus on multi-agent environments for smart building HVAC control, which inherently involve stochastic dynamics, distribution shifts, partial observability, and inter-agent coupling. These environments are particularly well-suited to test robustness, as the environmental models can vary across days and building locations (thus climate conditions). More details about the environment and the experiment setup are provided in Appendix F. Our code can be found at `https://github.com/crqu/robust-coMARL`.

**Evaluation protocol.** To assess generalization under distribution shift (i.e., model uncertainty), we adopt the following protocol. In the **training phase**, each algorithm is trained on a single environment. For robust MARL baselines that require multiple environments, we follow their standard protocol and train them on a fixed set of environments. In the **evaluation phase**, trained policies are deployed on unseen configurations that differ from those used in training, simulating realistic deployment where distribution shifts inevitably arise. This design allows us to explicitly measure robustness to changes in environment dynamics rather than simple memorization of training conditions.

**Baselines.** We compare our robust value factorization methods against:

- Non-robust factorization methods: VDN, QMIX, and QTRAN trained without robustness considerations, representing the standard CTDE paradigm.
- Existing robust CTDE baseline: the multi-agent group distributionally robust algorithm from Liu et al. (2025), which we refer to as "GroupDR". While the original work used only the VDN architecture, we extend the algorithm to QMIX and QTRAN for completeness.

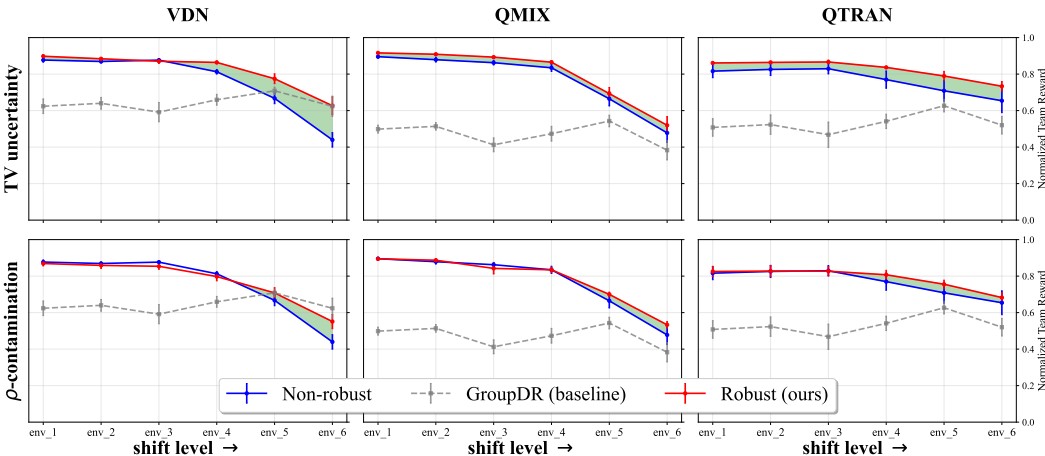

Figure 2: Normalized performance (averaged over 5 independent training runs, with error bars showing standard error) across different environment configurations for our robust MARL algorithms and other baselines. Each panel corresponds to one value factorization method. Robustness gain is the difference in reward (shaded area) between Robust (ours) and Non-robust, which shows the out-of-distribution performance improvement from the robust training.

**Experiment 1: climatic shifts.** We first test robustness under shifts induced by changes in climate conditions. Results (averaged over five seeds) are shown in Fig. 2. Our robust MARL algorithms consistently outperform both non-robust counterparts and the group DR baseline. Notably, performance degradation scales with the severity of the shift (e.g., `env_6` deviates most from the training environment, `env_1`), but our methods maintain relatively high returns. In contrast, the GroupDR baseline exhibits little sensitivity to shift severity, reflecting its reliance on worst-case rewards computed only from configurations encountered during training.

**Experiment 2: seasonal shifts.** We next evaluate robustness to seasonal shifts, training algorithms on `season_1` data and evaluating on `season_2`. Results are reported in Table 1, showing mean

and standard error of normalized episodic returns. The results show that robust value factorization algorithms with TV uncertainty set achieve consistent robustness gain against seasonal shifts.

| Factorization Methods | VDN | QMIX | QTRAN |
|---|---|---|---|
| Non-robust | $0.877 \pm 0.012$ | $0.895 \pm 0.008$ | $0.816 \pm 0.036$ |
| baseline (GroupDR) | $0.624 \pm 0.040$ | $0.499 \pm 0.022$ | $0.508 \pm 0.048$ |
| Robust (TV-uncertainty) | $\mathbf{0.898 \pm 0.008}$ | $\mathbf{0.916 \pm 0.006}$ | $\mathbf{0.861 \pm 0.006}$ |
| Robust ($\rho$-contamination) | $0.869 \pm 0.013$ | $\mathbf{0.911 \pm 0.005}$ | $\mathbf{0.825 \pm 0.028}$ |

Table 1: Final Performances under seasonal shifts for our robust MARL algorithms and other baselines (mean $\pm$ standard error over 5 independent training runs). Values outperforming both the non-robust and group DR baselines are highlighted in bold.

**Experiment 3: climatic and seasonal shifts.** Finally, we test on the most extreme case, where we have distribution shifts arising from climatic and seasonal shifts. The results are presented in Table 2, with our robust MARL algorithms achieving 10-40% higher average reward than the non-robust baseline. Notably, QTRAN-based robust MARL algorithms demonstrate strong out-of-distribution performance and stability.

| Factorization Methods | VDN | QMIX | QTRAN |
|---|---|---|---|
| Non-robust | $0.440 \pm 0.040$ | $0.478 \pm 0.052$ | $0.654 \pm 0.066$ |
| baseline (GroupDR) | $0.624 \pm 0.056$ | $0.383 \pm 0.053$ | $0.520 \pm 0.049$ |
| Robust (TV-uncertainty) | $\mathbf{0.627 \pm 0.049}$ | $\mathbf{0.520 \pm 0.048}$ | $\mathbf{0.733 \pm 0.026}$ |
| Robust ($\rho$-contamination) | $0.551 \pm 0.039$ | $\mathbf{0.500 \pm 0.075}$ | $\mathbf{0.682 \pm 0.026}$ |

Table 2: Final Performances under climatic and seasonal shifts for our robust MARL algorithms and other baselines (mean $\pm$ standard error over 5 independent training runs). Values outperforming both the non-robust and group DR baselines are highlighted in bold.

**Choice of $\rho$.** Theoretically, $\rho$ should be chosen based on prior estimation of the model uncertainty level. Practically, we select $\rho$ by training on `env_1` and validating on `env_2` and `env_3`, which yields stable performance without overfitting to a single shift.

**Robustness in cooperative MARL.** A noteworthy finding is that robustness in cooperative MARL does *not necessarily* entail reduced performance in the training environment. Unlike in single-agent robust RL, where conservatism often penalizes in-distribution returns, explicitly modeling robustness here mitigates errors from partial observability and decentralized execution. In several cases, robust training even improves in-distribution performance relative to non-robust baselines, suggesting that robustness can simultaneously enhance stability and adaptability in multi-agent systems.

## 5.2 STARCRAFT II

We additionally conduct experiments in SMAC (Samvelyan et al., 2019), a well-known benchmark consisting of two teams of agents engaged in cooperative combat scenarios based on StarCraft II. We focus on the hard `3s_vs_5z` map. In the test environment, we introduce distribution shift by adding noise to each agent's observation of every enemy unit's normalized position, sampled from $\mathcal{N}(0, 0.75^2)$. Since QTRAN has been shown to perform poorly on SMAC (Son et al., 2020), we report results for the $\rho$-contamination uncertainty set using only VDN and QMIX (averaged over five seeds) in Fig. 3. The results demonstrate that for small values of $\rho$, our robust MARL algorithms significantly improve out-of-distribution performance.

**Robustness parameter evaluation.** We further compare the final performance of our algorithms against their non-robust baselines, reporting the improvement in the test win rate for different choices of $\rho$ relative to the baseline. The results (averaged over five seeds) are shown in Fig. 4. Interestingly, the test win rate first increases as $\rho$ grows, and then decreases. This observation aligns with our theory: when $\rho$ is small relative to the shift level, explicitly modeling distribution shift during training yields improved out-of-distribution performance. However, when $\rho$ becomes large, the robust MARL algorithms become overly conservative, leading to degraded performance.

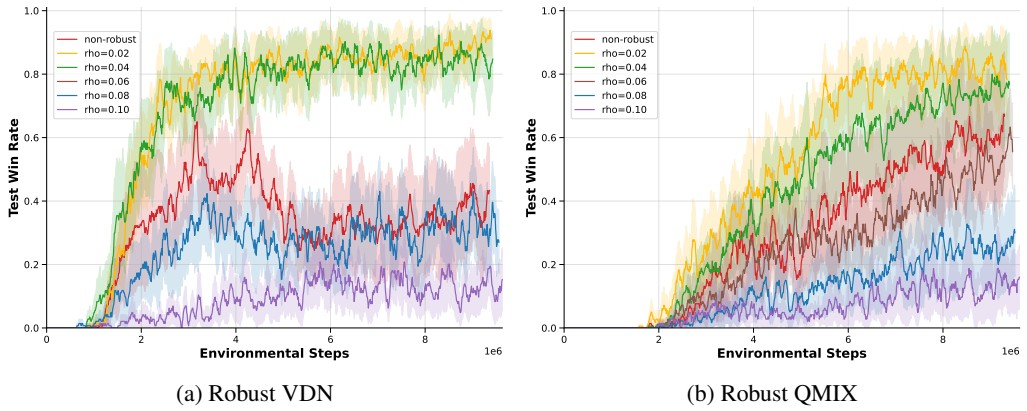

| (a) Robust VDN | (b) Robust QMIX |

Figure 3: Performance of our robust MARL algorithms and their non-robust baselines in SMAC (`3s_vs_5z` map). Each algorithm is evaluated every 10,000 environment steps, with each evaluation averaged over 32 episodes. Shaded regions denote the standard error across 5 random seeds. For small $\rho$, the robust algorithms significantly outperform their non-robust counterparts.

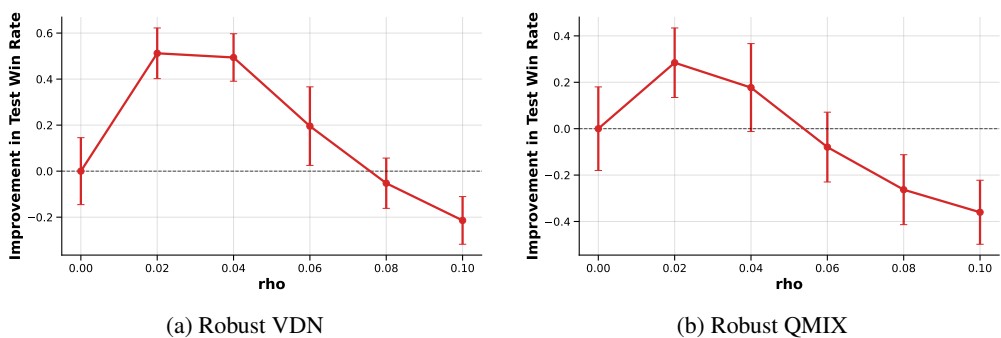

| (a) Robust VDN | (b) Robust QMIX |

Figure 4: Improvement in final test win rate of our robust MARL algorithms over their non-robust baselines in SMAC (`3s_vs_5z` map) for different values of $\rho$. Error bars denote the standard error across 5 random seeds.

## 6 CONCLUSION

In this work, we introduce Distributionally robust IGM (DrIGM), a robustness principle for cooperative MARL that extends the classical IGM property to settings with environmental uncertainties. Whereas naïvely "robustifying" individual agent policies fails to align robust individual policies with the joint robust policy, the DrIGM offers a principled framework for constructing robust individual action values that remain aligned with the joint robust policy, thereby enabling decentralized greedy execution under uncertainty.

Building on this foundation, we derive DrIGM-based robust value factorization algorithms for VDN, QMIX, and QTRAN, trained via robust Bellman operators under standard uncertainty sets ($\rho$-contamination and total variation). Empirically, on a high-fidelity building HVAC control benchmark, our methods consistently mitigate out-of-distribution performance degradation arising from climatic and seasonal shifts. On a StarCraft II game-playing benchmark, our methods likewise improve out-of-distribution performance under added observation noise. Unlike single-agent robust RL, where conservatism often harms in-distribution returns, we find that robustness in cooperative MARL can simultaneously enhance stability and adaptability.

While we introduced the DrIGM framework for a global uncertainty set, we believe it may be possible to further extend this framework. Future work includes developing DrIGM-compliant algorithms under agent-wise uncertainty sets and exploring additional training paradigms (e.g., decentralized training) to further broaden applicability.

## REPRODUCIBILITY STATEMENT

We release a github repository containing all code, configuration files, and scripts needed to reproduce our results, including data generation and figure plotting. All proofs for the main paper are stated in Appendix C. Algorithm pseudocode is also provided in Appendix E.

## ACKNOWLEDGMENT

KP acknowledges support from the 'PIMCO Postdoctoral Fellow in Data Science' fellowship at the California Institute of Technology. The major part of this work was done while KP was at the California Institute of Technology. EM acknowledges support from NSF award 2240110. This work acknowledges support from NSF CNS-2146814, CPS-2136197, CNS-2106403, NGSDI-2105648, and funding from the Resnick Sustainability Institute as well as a Quad Fellowship.

## AUTHOR CONTRIBUTIONS

**Chengrui Qu** led the technical development of the project, driving the theory and core methodology, running and analyzing the experiments, and taking a leading role in writing the paper.

**Christopher Yeh** co-developed the project and contributed in all aspects of experiments and writing.

**Kishan Panaganti** co-developed the project direction and prepared the initial proof of concept, advised CQ and CY, and played a significant role in paper writing.

**Eric Mazumdar** co-developed the project direction, advised CQ and KP.

**Adam Wierman** co-developed the project direction, advised CQ, CY, and KP, and played a significant role in paper writing.

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

# A    RELATED WORK

**Single-agent Distributionally Robust RL (DR-RL).**    The single-agent setting is typically formalized as a robust Markov decision process (MDP). A substantial literature studies finite-sample guarantees for distributionally robust RL, exploring a variety of ambiguity-set designs (Iyengar, 2005; Xu & Mannor, 2012; Wolff et al., 2012; Kaufman & Schaefer, 2013; Ho et al., 2018; Smirnova et al., 2019; Ho et al., 2021; Goyal & Grand-Clement, 2022; Derman & Mannor, 2020; Tamar et al., 2014; Panaganti & Kalathil, 2021a; Roy et al., 2017; Derman et al., 2018; Mankowitz et al., 2019). Most relevant to our work are tabular robust MDPs with $(s, a)$-rectangular uncertainty sets defined by total-variation balls (Yang et al., 2022; Panaganti & Kalathil, 2021b; Xu et al., 2023; Dong et al., 2022; Liu & Xu, 2024; Panaganti et al., 2022) or $\rho$-contamination models (Wang & Zou, 2022; Zhang et al., 2024), for which minimax dynamic programming and learning algorithms admit provable performance bounds.

**Value factorization methods for cooperative MARL.**    Value factorization is the standard mechanism for scalable cooperative MARL under CTDE. Early work adopts simple additivity (VDN (Sunehag et al., 2017)), while QMIX (Rashid et al., 2020) learn a state-conditioned monotone combiner to enlarge the function class without violating the IGM requirement. QTRAN (Son et al., 2019) further relax the monotonicity assumption with consistency constraints. Other approaches include attention-based mixers (e.g., QAtten (Yang et al., 2020), REFIL (Iqbal et al., 2021)), dueling-style decompositions (QPlex (Wang et al., 2021)) and residual designs (ResQ (Shen et al., 2022)). Building on this body of work, we develop robust value-factorization algorithms with provable robustness guarantees under model uncertainty, enabling robust decentralized execution in partially observable cooperative settings.

**Robustness in MARL.**    In general MARL, robustness is typically studied within Markov games, where uncertainty can be modeled in different components, such as the state space (Han et al., 2022; He et al., 2023; Zhou & Liu, 2023; Zhang et al., 2023), other agents (Li et al., 2019; Kannan et al., 2023), and environmental dynamics (Zhang et al., 2020a; Liu et al., 2025). We refer readers to Vial et al. (2022) for an overview. This work considers robustness to model uncertainty, primarily studied via distributionally robust optimization (DRO) (Rahimian & Mehrotra, 2019; Gao, 2020; Bertsimas et al., 2018; Duchi & Namkoong, 2018; Blanchet & Murthy, 2019; Fonseca & Junca, 2023; Qu et al., 2025; Mohajerin Esfahani & Kuhn, 2018; Noyan et al., 2022), where most prior efforts target Nash equilibria and provide provable (actor–critic / Q-learning) algorithms (Zhang et al., 2020a; Kardeş et al., 2011; Ma et al., 2023; Blanchet et al., 2023; Shi et al., 2024; Liu et al., 2025), often under full observability or individually rewarded settings. We complement this line by addressing the cooperative, partially observable CTDE regime, where agents receive a single joint reward and act only local observations.

In cooperative MARL, robustness has been modeled along several complementary axes, including adversarial (Byzantine) teammates (Li et al., 2024), state/observation disturbances (Guo et al., 2024), communication errors (Yu et al., 2024), risk-sensitive objectives that guard against tail events under a fixed model (Shen et al., 2023), and explicit model uncertainty Kwak et al. (2010); Zhang et al. (2020b). Focusing on the last category, Kwak et al. (2010) address model uncertainty with sparse, execution-time communication, whereas Zhang et al. (2020b) study settings in which each agent observes the full state and receives individual reward. Similarly, Bukharin et al. (2023) also considers settings where each agent receives individual reward, and they achieve robustness by controlling the Lipschitz constant of each agent's policy. In contrast, our work targets robustness to model uncertainty in the cooperative CTDE setting, complementing prior approaches by providing a systematic framework that does *not* require real-time communication and operates under partial observability with a single team reward.

# B    EXAMPLES

**Example 1** (Naïve single-agent robust action values cannot guarantee DrIGM)**.** *Consider a robust cooperative two-agent task (illustrated in Fig. 5) with action spaces $\mathcal{A}_1 = \mathcal{A}_2 = \{1, 2\}$, state space $\mathcal{S} = \{s_0, s_1, s_2, s_3, s_4\}$, and uncertainty set $\mathcal{P} = \{P_1, P_2\}$. Let $s_0$ be the initial state, and let $s_1, s_2, s_3$ and $s_4$ all be absorbing states with zero reward. We assume each agent observes the full state. For $P_1$, all the transitions are fully deterministic. $P_2$ differs from $P_1$ only in transitions on joint*

*actions* $(1,2)$ *and* $(2,1)$, *given by:*

$$\mathbb{P}(S_2 \mid 1,2) = \frac{1}{3}, \mathbb{P}(S_3 \mid 1,2) = \frac{2}{3},$$

$$\mathbb{P}(S_2 \mid 2,1) = \frac{2}{3}, \mathbb{P}(S_3 \mid 2,1) = \frac{1}{3},$$

*As shown in Fig. 5, the optimal joint action value function at state $s_0$ is (we omit the $\gamma/(1-\gamma)$ factor for clarity)*

$$Q_{\text{tot}}^{P_1}(s_0,1,1) = 0.7, \quad Q_{\text{tot}}^{P_1}(s_0,1,2) = 0.4, \quad Q_{\text{tot}}^{P_1}(s_0,2,1) = 1.0, \quad Q_{\text{tot}}^{P_1}(s_0,2,2) = 0.7;$$

$$Q_{\text{tot}}^{P_2}(s_0,1,1) = 0.7, \quad Q_{\text{tot}}^{P_2}(s_0,1,2) = 0.8, \quad Q_{\text{tot}}^{P_2}(s_0,2,1) = 0.6, \quad Q_{\text{tot}}^{P_2}(s_0,2,2) = 0.7.$$

*Therefore, the robust joint action-value function $Q_{\text{tot}}^{\mathcal{P}}(s,\mathbf{a}) = \inf_{P \in \mathcal{P}} Q_{\text{tot}}^{P}(s,\mathbf{a})$ is given by:*

$$Q_{\text{tot}}^{\mathcal{P}}(s_0,1,1) = 0.7, \quad Q_{\text{tot}}^{\mathcal{P}}(s_0,1,2) = 0.4, \quad Q_{\text{tot}}^{\mathcal{P}}(s_0,2,1) = 0.6, \quad Q_{\text{tot}}^{\mathcal{P}}(s_0,2,2) = 0.7,$$

*It is straightforward to check that the following individual action-value functions $\{Q_i^{P_j}\}_{i,j \in [2]}$ satisfy $Q_{\text{tot}}^{P}(s,a_1,a_2) = Q_1^{P}(s,a_1) + Q_2^{P}(s,a_2)$ for all $s \in \mathcal{S}$ and $P \in \mathcal{P}$, which is a special case of the classical IGM property:*

$$Q_1^{P_1}(s_0,1) = 0, \quad Q_1^{P_1}(s_0,2) = 0.3, \quad Q_2^{P_1}(s_0,1) = 0.7, \quad Q_2^{P_1}(s_0,2) = 0.4,$$

$$Q_1^{P_2}(s_0,1) = 0.2, \quad Q_1^{P_2}(s_0,2) = 0.1, \quad Q_2^{P_2}(s_0,1) = 0.5, \quad Q_2^{P_2}(s_0,2) = 0.6.$$

*Suppose the robust individual action-value function is defined as $Q_i^{\text{rob}}(s,a) = \inf_{P \in \mathcal{P}} Q_i^{P}(s,a)$, as in the single-agent DR-RL literature. Therefore, the robust individual action-value functions are given by:*

$$Q_1^{\text{rob}}(s_0,1) = 0, \quad Q_1^{\text{rob}}(s_0,2) = 0.1, \quad Q_2^{\text{rob}}(s_0,1) = 0.5, \quad Q_2^{\text{rob}}(s_0,2) = 0.4,$$

*At $s_0$, these robustifications fail to satisfy DrIGM:*

$$(2,1) = \left( \arg\max_{a_1} Q_1^{\text{rob}}(s_0,a_1), \arg\max_{a_2} Q_2^{\text{rob}}(s_0,a_2) \right) \notin \arg\max_{\mathbf{a}} Q_{\text{tot}}^{\mathcal{P}}(s_0,\mathbf{a}) = \{(1,1),(2,2)\}.$$

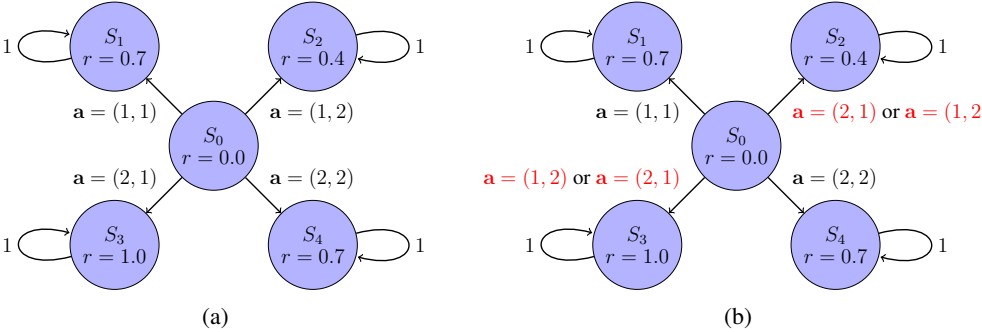

Figure 5: Fig. 5a is the MDP under transition kernel $P_1$, Fig. 5b is under $P_2$. The two differ in their transition probabilities to $s_2$ and $s_3$.

**Example 2** (DrIGM can address cases where IGM fails.)**.** *Consider a similar robust cooperative two-agent task (illustrated in Fig. 6) with action spaces $\mathcal{A}_1 = \mathcal{A}_2 = \{1,2\}$, state space $\mathcal{S} = \{s_0,s_1,s_2,s_3,s_4\}$, and uncertainty set $\mathcal{P} = \{P_1,P_2\}$. Let $P_1$ be the training and testing environment. For $P_1$, all the transitions are fully deterministic, the optimal joint action value function at state $s_0$ is (we omit the $\gamma/(1-\gamma)$ factor for clarity):*

$$Q_{\text{tot}}^{P_1}(s_0,1,1) = 0.7, \; Q_{\text{tot}}^{P_1}(s_0,1,2) = 0.4, \quad Q_{\text{tot}}^{P_1}(s_0,2,1) = 1.0, \; Q_{\text{tot}}^{P_1}(s_0,2,2) = 0.5.$$

*$P_2$ differs from $P_1$ in that all actions leads to $S_4$. Therefore, the optimal joint action value function at state $s_0$ is (we omit the $\gamma/(1-\gamma)$ factor for clarity):*

$$Q_{\text{tot}}^{P_2}(s_0,1,1) = 0.4, \; Q_{\text{tot}}^{P_2}(s_0,1,2) = 0.4, \quad Q_{\text{tot}}^{P_2}(s_0,2,1) = 0.4, \; Q_{\text{tot}}^{P_2}(s_0,2,2) = 0.4.$$

*It can be verified that $P_1$ does not admit a VDN-style value decomposition, but the worst case, $P_2$, admits a feasible VDN-style value decomposition.*

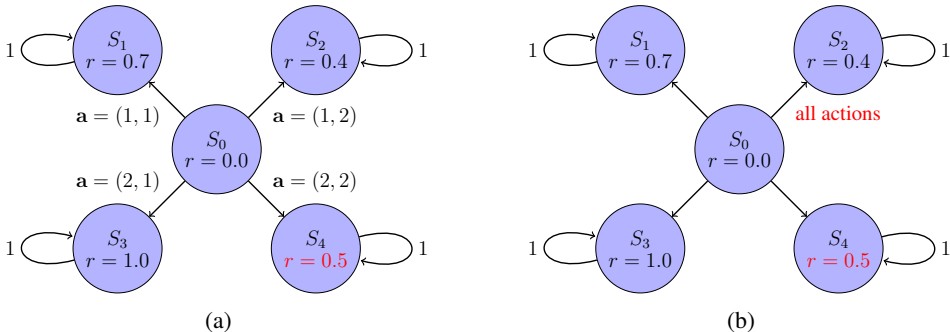

Figure 6: Fig. 6a is the MDP under transition kernel $P_1$, Fig. 6b is under $P_2$.

## C   PROOFS

### C.1   PROOF OF THEOREM 1

*Proof.* Recall that for all $\mathbf{a} \in \mathcal{A}$, we have

$$Q_{\text{tot}}^{\mathcal{P}}(\mathbf{h}, \mathbf{a}) = \inf_{P \in \mathcal{P}} Q_{\text{tot}}^{P}(\mathbf{h}, \mathbf{a}) \qquad \text{(Eq. (3))}$$

$$P^{\text{worst}}(\mathbf{h}, \mathbf{a}) \in \arg \inf_{P \in \mathcal{P}} Q_{\text{tot}}^{P}(\mathbf{h}, \mathbf{a}). \qquad \text{(Equation (4))}$$

Thus, $Q_{\text{tot}}^{P^{\text{worst}}(\mathbf{h}, \mathbf{a})}(\mathbf{h}, \mathbf{a}) = Q_{\text{tot}}^{\mathcal{P}}(\mathbf{h}, \mathbf{a})$. In Equation (4), we also defined $\bar{\mathbf{a}} \in \arg\max_{\mathbf{a} \in \mathcal{A}} Q_{\text{tot}}^{\mathcal{P}}(\mathbf{h}, \mathbf{a})$. Since $P^{\text{worst}}(\mathbf{h}, \bar{\mathbf{a}}) \in \mathcal{P}$, by assumption there exist $[Q_i^{P^{\text{worst}}(\mathbf{h}, \bar{\mathbf{a}})}]_{i \in [N]}$ that satisfy IGM for $Q_{\text{tot}}^{P^{\text{worst}}(\mathbf{h}, \bar{\mathbf{a}})}$ under $\mathbf{h}$. Therefore,

$$\left( \arg\max_{a_1} Q_1^{\text{rob}}(h_1, a_1), \ldots, \arg\max_{a_N} Q_N^{\text{rob}}(h_N, a_N) \right)$$

$$= \left( \arg\max_{a_1} Q_1^{P^{\text{worst}}(\mathbf{h}, \bar{\mathbf{a}})}(h_1, a_1), \ldots, \arg\max_{a_N} Q_N^{P^{\text{worst}}(\mathbf{h}, \bar{\mathbf{a}})}(h_N, a_N) \right) \qquad \text{(Eq. (4))}$$

$$\subseteq \arg\max_{\mathbf{a}} Q_{\text{tot}}^{P^{\text{worst}}(\mathbf{h}, \bar{\mathbf{a}})}(\mathbf{h}, \mathbf{a}) \qquad \text{(IGM)}$$

$$\subseteq \arg\max_{\mathbf{a}} Q_{\text{tot}}^{P^{\text{worst}}(\mathbf{h}, \mathbf{a})}(\mathbf{h}, \mathbf{a}) \qquad \text{(Eq. (4))}$$

$$= \arg\max_{\mathbf{a}} Q_{\text{tot}}^{\mathcal{P}}(\mathbf{h}, \mathbf{a}),$$

which shows that $[Q_i^{\text{rob}}]_{i \in [N]}$ satisfy DrIGM under $\mathbf{h}$. □

**Remark 1.** *The statement of Theorem 1 assumes that for all $P \in \mathcal{P}$ there exist $[Q_i^P]_{i \in [N]}$ satisfying IGM for $Q_{\text{tot}}^P$ under joint history $\mathbf{h} \in \mathcal{H}$. However, we note here that this assumption can be relaxed to only requiring that there exist $[Q_i]_{i \in [N]}$ satisfying IGM for $Q_{\text{tot}}^{P^{\text{worst}}}$ under $\mathbf{h}$.*

**Remark 2.** *As shown in Example 1, an adversarial model $P \in \mathcal{P}$ that minimizes one agent's value need not coincide with the adversarial model $P' \in \mathcal{P}$ that minimizes the joint value. Theorem 1 circumvents this problem by directly considering the global worst case, i.e.,*

$$Q_i^{\text{rob}}(h_i, a_i) := Q_i^{P^{\text{worst}}(\mathbf{h}, \bar{\mathbf{a}})}(h_i, a_i). \qquad (18)$$

*Given this definition, a robust joint action is given by $\bar{\mathbf{a}} = (1, 1)$ or $(2, 2)$, and there exist robust individual action-value functions are given by:*

$$Q_1^{\text{rob}}(s_0, 1) = 0.2, \qquad Q_1^{\text{rob}}(s_0, 2) = 0.1, \quad Q_2^{\text{rob}}(s_0, 1) = 0.7, \qquad Q_2^{\text{rob}}(s_0, 2) = 0.4,$$

*where the robust individual action is given by $a_1 = 1, a_2 = 1$. Alternatively, another set of individual action-value functions are given by:*

$$Q_1^{\text{rob}}(s_0, 1) = 0, \qquad Q_1^{\text{rob}}(s_0, 2) = 0.3, \quad Q_2^{\text{rob}}(s_0, 1) = 0.5, \qquad Q_2^{\text{rob}}(s_0, 2) = 0.6,$$

*where the robust individual action is given by $a_1 = 2, a_2 = 2$. Either way, the robust individual actions are aligned with the joint actions.*

## C.2 PROOF OF THEOREM 2

*Proof.* We proceed by proving that IGM holds for under each of the three conditions given in Theorem 2. By Theorem 1, this suffices to show that DrIGM holds for under each of the three conditions given in Theorem 2.

**VDN condition.** Given a joint history $\mathbf{h} \in \mathcal{H}$, for any $P \in \mathcal{P}$, we have

$$Q_{\text{tot}}^P(\mathbf{h}, \mathbf{a}) = \sum_{i \in [N]} Q_i^P(h_i, a_i), \quad \forall \mathbf{a} = (a_1, \ldots, a_N) \in \mathcal{A}.$$

Let $\bar{a}_i = \arg\max_{a_i} Q_i^P(h_i, a_i)$ for $i \in [N]$, and let $\bar{\mathbf{a}} = [\bar{a}_i]_{i \in [N]}$. Then, for any $\mathbf{a} \in \mathcal{A}$,

$$\begin{aligned} Q_{\text{tot}}^P(\mathbf{h}, \mathbf{a}) &= \sum_{i \in [N]} Q_i^P(h_i, a_i), \\ &\leq \sum_{i \in [N]} Q_i^P(h_i, \bar{a}_i) \qquad \text{(Definition of } \bar{a}_i) \\ &= Q_{\text{tot}}^P(\mathbf{h}, \bar{\mathbf{a}}). \end{aligned}$$

This implies that

$$\left( \arg\max_{a_1} Q_1^P(h_1, a_1), \ldots, \arg\max_{a_N} Q_N^P(h_N, a_N) \right) \subseteq \arg\max_{\mathbf{a}} Q_{\text{tot}}^P(\mathbf{h}, \mathbf{a}),$$

so $[Q_i^P]_{i \in [N]}$ satisfy IGM for $Q_{\text{tot}}^P$ under $\mathbf{h}$. Therefore, by Theorem 1, $[Q_i^{\text{rob}}]_{i \in [N]}$ satisfy DrIGM for $Q_{\text{tot}}^{\mathcal{P}}$ under $\mathbf{h}$.

**QMIX condition.** Given a joint history $\mathbf{h} \in \mathcal{H}$, for any $P \in \mathcal{P}$, suppose the following monotonicity property holds:

$$\frac{\partial Q_{\text{tot}}^P(\mathbf{h}, \mathbf{a})}{\partial Q_i^P(h_i, a_i)} \geq 0, \quad \forall i \in [N], \ \mathbf{a} = (a_1, \ldots, a_N) \in \mathcal{A}.$$

Let $\bar{a}_i = \arg\max_{a_i} Q_i^P(h_i, a_i)$ for $i \in [N]$, and let $\bar{\mathbf{a}} = [\bar{a}_i]_{i \in [N]}$. Given that $\partial Q_{\text{tot}}^P(\mathbf{h}, \mathbf{a})/\partial Q_1^P(h_1, a_1) \geq 0$ and $\bar{a}_1 = \arg\max_{a_1} Q_1^P(h_1, a_1)$, we have (for any $\mathbf{a} \in \mathcal{A}$)

$$Q_{\text{tot}}^P(\mathbf{h}, \mathbf{a}) \leq Q_{\text{tot}}^P(\mathbf{h}, \bar{a}_1, a_2, \ldots, a_N).$$

Applying the same logic to all $i \in [N]$ yields that for any $\mathbf{a} \in \mathcal{A}$,

$$Q_{\text{tot}}^P(\mathbf{h}, \mathbf{a}) \leq Q_{\text{tot}}^P(\mathbf{h}, \bar{a}_1, \bar{a}_2, \ldots, \bar{a}_N) \tag{19}$$
$$= Q_{\text{tot}}^P(\mathbf{h}, \bar{\mathbf{a}}). \tag{20}$$

This implies that

$$\left( \arg\max_{a_1} Q_1^P(h_1, a_1), \ldots, \arg\max_{a_N} Q_N^P(h_N, a_N) \right) \subseteq \arg\max_{\mathbf{a}} Q_{\text{tot}}^P(\mathbf{h}, \mathbf{a}),$$

so $[Q_i^P]_{i \in [N]}$ satisfy IGM for $Q_{\text{tot}}^P$ under $\mathbf{h}$. Therefore, by Theorem 1, $[Q_i^{\text{rob}}]_{i \in [N]}$ satisfy DrIGM for $Q_{\text{tot}}^{\mathcal{P}}$ under $\mathbf{h}$.

**QTRAN condition.** Given a joint history $\mathbf{h} \in \mathcal{H}$, for any $P \in \mathcal{P}$, we have (for all $\mathbf{a} = (a_1, \ldots, a_N) \in \mathcal{A}$):

$$\sum_{i=1}^N Q_i^P(h_i, a_i) - Q_{\text{tot}}^P(\mathbf{h}, \mathbf{a}) + V_{\text{tot}}(\mathbf{h}) = \begin{cases} 0, & \mathbf{a} = \bar{\mathbf{a}}, \\ \geq 0, & \mathbf{a} \neq \bar{\mathbf{a}}, \end{cases} \tag{21}$$

where $\bar{\mathbf{a}} = [\bar{a}_i]_{i \in [N]}$ with $\bar{a}_i = \arg\max_{a_i} Q_i^P(h_i, a_i)$ and $V_{\text{tot}}(\mathbf{h}) = \max_{\mathbf{a}} Q_{\text{tot}}^P(\mathbf{h}, \mathbf{a}) - \sum_{i=1}^N Q_i^P(h_i, a_i)$. Therefore,

$$Q_{\text{tot}}^P(\mathbf{h}, \bar{\mathbf{a}}) = \sum_{i=1}^N Q_i^P(h_i, \bar{a}_i) + V_{\text{tot}}(\mathbf{h}) \qquad \text{(Eq. (21))}$$

$$= \max_{\mathbf{a}} Q^P_{\text{tot}}(\mathbf{h}, \mathbf{a}), \qquad\qquad\qquad (\text{Definition of } V(\mathbf{h}))$$

This implies that

$$\left( \arg\max_{a_1} Q^P_1(h_1, a_1), \ldots, \arg\max_{a_N} Q^P_N(h_N, a_N) \right) \subseteq \arg\max_{\mathbf{a}} Q^P_{\text{tot}}(\mathbf{h}, \mathbf{a}),$$

so $[Q^P_i]_{i \in [N]}$ satisfy IGM for $Q^P_{\text{tot}}$ under $\mathbf{h}$. Therefore, by Theorem 1, $[Q^{\text{rob}}_i]_{i \in [N]}$ satisfy DrIGM for $Q^{\mathcal{P}}_{\text{tot}}$ under $\mathbf{h}$.

Combining the three cases concludes the proof of Theorem 2. $\qquad\qquad\qquad\qquad\qquad\qquad\square$

### C.3 PROOF OF THEOREM 3

*Proof.* Recall that given an uncertainty set $\mathcal{P}$, the robust joint action value is defined as,

$$Q^{\mathcal{P}}_{\text{tot}}(\mathbf{h}, \mathbf{a}) := \inf_{P \in \mathcal{P}} Q^P_{\text{tot}}(\mathbf{h}, \mathbf{a}), \ \forall (\mathbf{h}, \mathbf{a}) \in \mathcal{H} \times \mathcal{A}. \qquad\qquad (22)$$

Given that $P_{\text{test}} \in \mathcal{P}$, we directly have:

$$Q^{\mathcal{P}}_{\text{tot}}(\mathbf{h}, \mathbf{a}) \leq Q^{P_{\text{test}}}_{\text{tot}}(\mathbf{h}, \mathbf{a}), \ \forall \mathbf{h} \in \mathcal{H}, \ \mathbf{a} \in \mathcal{A}. \qquad\qquad (23)$$

This concludes the proof. $\qquad\qquad\qquad\qquad\qquad\qquad\qquad\qquad\qquad\qquad\qquad\square$

## D ROBUST BELLMAN OPERATORS

We start by introducing the assumptions needed to derive the robust bellman operators.

**Assumption 1** (Fail-state (Panaganti et al., 2022))**.** *The robust Dec-POMDP has a* fail state $s_f$ *such that*

$$r(s_f, \mathbf{a}) = 0 \quad and \quad P_{s_f, \mathbf{a}}(s_f) = 1, \qquad \forall \mathbf{a} \in \mathcal{A}, \ \forall P \in \mathcal{P}. \qquad (24)$$

This requirement is mild, as fail states naturally arise in both simulated and physical systems. For example, in robotics, a configuration where the robot falls and cannot recover, whether in simulators such as MuJoCo or in real hardware, serves as a natural fail state. We can further relax it to the following assumption.

**Assumption 2** (Vanishing minimal value (Lu et al., 2024))**.** *The underlying RMDP satisfies*

$$\min_{s \in \mathcal{S}} V^{\mathcal{P}}_{\text{tot}}(s) = 0. \qquad\qquad\qquad\qquad (25)$$

*Without loss of generality, we also assume that any initial state* $s_1 \notin \arg\min_{s \in \mathcal{S}} V^{\mathcal{P}}_{\text{tot}}(s)$.

This assumption states that the lowest achievable robust value across all states is normalized to zero. The exclusion of the minimizing state as the starting point rules out the degenerate case where the agent begins with zero guaranteed return.

$\rho$**-contamination uncertainty set.** Given a $\rho$-contamination uncertainty set defined in Eq. (7), the robust bellman operator can be expanded as:

$$(\mathcal{T}^{\mathcal{P}} Q)(\mathbf{h}, \mathbf{a}) = r(s, \mathbf{a}) + \gamma \inf_{P_{\mathbf{h}, \mathbf{a}} \in \mathcal{P}_{\mathbf{h}, \mathbf{a}}} \mathbb{E}_{\mathbf{h}' \sim P_{\mathbf{h}, \mathbf{a}}} \left[ \max_{\mathbf{a}' \in \mathcal{A}} Q(\mathbf{h}', \mathbf{a}') \right]. \qquad (26)$$

$$= r(s, \mathbf{a}) + \gamma(1 - \rho) \, \mathbb{E}_{\mathbf{h}' \sim P^0_{\mathbf{h}, \mathbf{a}}} \left[ \max_{\mathbf{a}' \in \mathcal{A}} Q(\mathbf{h}', \mathbf{a}') \right] \qquad (27)$$

$$+ \rho \min_{s' \in \mathcal{S}} V^{\mathcal{P}}_{\text{tot}}(s'). \qquad\qquad\qquad (28)$$

Under Assumption 1 (Assumption 2), we obtain that:

$$(\mathcal{T}^{\mathcal{P}} Q)(\mathbf{h}, \mathbf{a}) = r(s, \mathbf{a}) + \gamma(1 - \rho) \, \mathbb{E}_{\mathbf{h}' \sim P^0_{\mathbf{h}, \mathbf{a}}} \left[ \max_{\mathbf{a}' \in \mathcal{A}} Q(\mathbf{h}', \mathbf{a}') \right] \quad (\text{Assumption 1 (Assumption 2)})$$

$$= r(s, \mathbf{a}) + \gamma(1 - \rho) \, \mathbb{E}_{\mathbf{h}' \sim P^0_{\mathbf{h}, \mathbf{a}}} \left[ Q(\mathbf{h}', \bar{\mathbf{a}}') \right], \qquad\qquad (\text{Definition 2})$$

where $\bar{a}'_i = \arg\max_{a'_i} Q^{\text{rob}}_i(h'_i, a'_i)$.

**TV-uncertainty set.** Leveraging Panaganti et al. (2022)[Proposition 1], given a TV-uncertainty set defined in Eq. (9), the robust bellman operator can be expanded as:

$$(\mathcal{T}Q_{\text{tot}}^{\mathcal{P}})(\mathbf{h}, \mathbf{a}) = r(s, \mathbf{a}) - \inf_{\eta \in [0, \frac{2}{\rho(1-\gamma)}]} \gamma \, \mathbb{E}_{\mathbf{h}' \sim P_{\mathbf{h},\mathbf{a}}^0} \left( \rho \left[ \eta(s, \mathbf{a}) - \inf_{s'' \in \mathcal{S}} V_{\text{tot}}^{\mathcal{P}}(s'') \right]_+ - \eta \right.$$
$$\left. + \left[ \eta(s, \mathbf{a}) - \max_{\mathbf{a}' \in \mathcal{A}} Q_{\text{tot}}^{\mathcal{P}}(\mathbf{h}', \mathbf{a}') \right]_+ \right). \tag{29}$$

Under Assumption 1 (Assumption 2), we obtain that:

$$(\mathcal{T}Q_{\text{tot}}^{\mathcal{P}})(\mathbf{h}, \mathbf{a}) = r(s, \mathbf{a}) - \inf_{\eta \in [0, \frac{2}{\rho(1-\gamma)}]} \gamma \, \mathbb{E}_{\mathbf{h}' \sim P_{\mathbf{h},\mathbf{a}}^0} \left( -(1-\rho)\eta(s, \mathbf{a}) \right.$$
$$\left. + \left[ \eta(s, \mathbf{a}) - \max_{\mathbf{a}' \in \mathcal{A}} Q_{\text{tot}}^{\mathcal{P}}(\mathbf{h}', \mathbf{a}') \right]_+ \right). \qquad \text{(Assumption 1 (Assumption 2))}$$

$$= r(s, \mathbf{a}) - \inf_{\eta \in [0, \frac{2}{\rho(1-\gamma)}]} \gamma \, \mathbb{E}_{\mathbf{h}' \sim P_{\mathbf{h},\mathbf{a}}^0} \left( -(1-\rho)\eta(s, \mathbf{a}) \right.$$
$$\left. + \left[ \eta(s, \mathbf{a}) - \max_{\mathbf{a}' \in \mathcal{A}} Q_{\text{tot}}^{\mathcal{P}}(h_1', \ldots, h_N', \bar{a}_1', \ldots, \bar{a}_N') \right]_+ \right). \qquad \text{(Definition 2)}$$

# E  ALGORITHMS

We offer a full description of our algorithms in this section, presented in Algorithms 2 to 7.

---

**Algorithm 2** Robust VDN with $\rho$-contamination uncertainty set

1: Input robustness parameter $\rho$, target network update frequency $f$ and $\varepsilon$
2: Initialize replay buffer $\mathcal{D}$
3: Initialize $[Q_i^{\text{rob}}]_{i \in [N]}$ with random parameters $\theta$
4: Initialize target parameters $\theta^- = \theta$
5: **for** episode $e = 1, \ldots, E$ **do**
6:     Observe initial state $s^0$ and observation $o_i^0 = \sigma_i(s^0)$ for each agent $i$.
7:     **for** $t = 1, \ldots, T$ **do**
8:         Each agent $i$ choose its action $a_i^t$ using $\varepsilon$-greedy policy.
9:         Take joint action $\mathbf{a}^t$, observe the next state $s^{t+1}$, reward $r^t$ and observation $o_i^{t+1} = \sigma_i(s^{t+1})$ for each agent $i$
10:       Store transition $(\mathbf{h}^t, \mathbf{a}^t, r^t, \mathbf{h}^{t+1})$ in replay buffer $\mathcal{D}$
11:       Sample a mini-batch of transitions $(\mathbf{h}, \mathbf{a}, r, \mathbf{h}')$ from $\mathcal{D}$
12:       Set $\bar{\mathbf{a}}' = [\arg\max_{a_i'} Q_i^{\text{rob}}(h_i', a_i'; \theta^-)]_{i \in [N]}$
13:       Set $Q_{\text{tot}}^{\mathcal{P},\text{VDN}}(\mathbf{h}, \mathbf{a}; \theta) = \sum_{i \in [N]} Q_i^{\text{rob}}(h_i, a_i; \theta)$
14:       Set $Q_{\text{tot}}^{\mathcal{P},\text{VDN}}(\mathbf{h}', \bar{\mathbf{a}}'; \theta^-) = \sum_{i \in [N]} Q_i^{\text{rob}}(h_i', \bar{a}_i'; \theta^-)$
15:       Set $y^{\text{target}} = r + \gamma(1-\rho)Q_{\text{tot}}^{\mathcal{P},\text{VDN}}(\mathbf{h}', \bar{\mathbf{a}}'; \theta^-)$
16:       Calculate TD loss $L_{\text{TD}} = (Q_{\text{tot}}^{\mathcal{P},\text{VDN}}(\mathbf{h}, \mathbf{a}; \theta) - y^{\text{target}})^2$
17:       Update $\theta$ by minimizing $L_{\text{TD}}$
18:       Update $\theta^- = \theta$ with frequency $f$
19:     **end for**
20: **end for**

---

# F  EXPERIMENT DETAILS

## F.1  TASK DESCRIPTION

We test our algorithms and baseline algorithms in `BuildingEnv` in Yeh et al. (2023). This environment considers the control of the heat flow in a multi-zone building so as to maintain a desired temperature setpoint. Building temperature simulation uses f*irst-principled physics models*, to capture the real-world dynamics. The environmental model and reward functions can differ from three climate types and locations (San Diego, Tucson, New York), which jointly decide the climate.

---

**Algorithm 3** Robust QMIX with $\rho$-contamination uncertainty set

---

1: Input robustness parameter $\rho$, target network update frequency $f$ and $\varepsilon$
2: Initialize replay buffer $\mathcal{D}$
3: Initialize $[Q_i^{\mathrm{rob}}]_{i \in [N]}$ and mixing network $f_\theta$ with random parameters $\theta$
4: Initialize target parameters $\theta^- = \theta$
5: **for** episode $e = 1, \ldots, E$ **do**
6:     Observe initial state $s^0$ and observation $o_i^0 = \sigma_i(s^0)$ for each agent $i$.
7:     **for** $t = 1, \ldots, T$ **do**
8:         Each agent $i$ choose its action $a_i^t$ using $\varepsilon$-greedy policy.
9:         Take joint action $\mathbf{a}^t$, observe the next state $s^{t+1}$, reward $r^t$ and observation $o_i^{t+1} = \sigma_i(s^{t+1})$
            for each agent $i$
10:         Store transition $(\mathbf{h}^t, \mathbf{a}^t, s^t, r^t, \mathbf{h}^{t+1}, s^{t+1})$ in replay buffer $\mathcal{D}$
11:         Sample a mini-batch of transitions $(\mathbf{h}, \mathbf{a}, s, r, \mathbf{h}', s')$ from $\mathcal{D}$
12:         Set $\bar{\mathbf{a}}' = [\arg\max_{a_i'} Q_i^{\mathrm{rob}}(h_i', a_i'; \theta^-)]_{i \in [N]}$
13:         Set $Q_{\mathrm{tot}}^{\mathcal{P},\mathrm{QMIX}}(\mathbf{h}, \mathbf{a}; \theta) = f_\theta((Q_i^{\mathrm{rob}}(h_i, a_i; \theta))_{i \in [N]}, s)$
14:         Set $Q_{\mathrm{tot}}^{\mathcal{P},\mathrm{QMIX}}(\mathbf{h}', \bar{\mathbf{a}}'; \theta^-) = f_{\theta^-}((Q_i^{\mathrm{rob}}(h_i', \bar{a}_i'; \theta^-))_{i \in [N]}, s')$
15:         Set $y^{\mathrm{target}} = r + \gamma(1 - \rho)Q_{\mathrm{tot}}^{\mathcal{P},\mathrm{QMIX}}(\mathbf{h}', \bar{\mathbf{a}}'; \theta^-)$
16:         Calculate TD loss $L_{\mathrm{TD}} = (Q_{\mathrm{tot}}^{\mathcal{P},\mathrm{QMIX}}(\mathbf{h}, \mathbf{a}; \theta) - y^{\mathrm{target}})^2$
17:         Update $\theta$ by minimizing $L_{\mathrm{TD}}$
18:         Update $\theta^- = \theta$ with frequency $f$
19:     **end for**
20: **end for**

---

**Algorithm 4** Robust QTRAN with $\rho$-contamination uncertainty set

---

1: Input robustness parameter $\rho$, target network update frequency $f$ and $\varepsilon$
2: Initialize replay buffer $\mathcal{D}$
3: Initialize $[Q_i^{\mathrm{rob}}]_{i \in [N]}$, $Q_{\mathrm{tot}}^{\mathcal{P},\mathrm{QTRAN}}$ and $V_{\mathrm{tot}}^{\mathcal{P},\mathrm{QTRAN}}$ with random parameters $\theta$
4: Initialize target parameters $\theta^- = \theta$
5: **for** episode $e = 1, \ldots, E$ **do**
6:     Observe initial state $s^0$ and observation $o_i^0 = \sigma_i(s^0)$ for each agent $i$.
7:     **for** $t = 1, \ldots, T$ **do**
8:         Each agent $i$ choose its action $a_i^t$ using $\varepsilon$-greedy policy.
9:         Take joint action $\mathbf{a}^t$, observe the next state $s^{t+1}$, reward $r^t$ and observation $o_i^{t+1} = \sigma_i(s^{t+1})$
            for each agent $i$
10:         Store transition $(\mathbf{h}^t, \mathbf{a}^t, r^t, \mathbf{h}^{t+1})$ in replay buffer $\mathcal{D}$
11:         Sample a mini-batch of transitions $(\mathbf{h}, \mathbf{a}, r, \mathbf{h}')$ from $\mathcal{D}$
12:         Set $\bar{\mathbf{a}}' = [\arg\max_{a_i'} Q_i^{\mathrm{rob}}(h_i', a_i'; \theta^-)]_{i \in [N]}$
13:         Set $y^{\mathrm{target}} = r + \gamma(1 - \rho)Q_{\mathrm{tot}}^{\mathcal{P},\mathrm{QTRAN}}(\mathbf{h}', \bar{\mathbf{a}}'; \theta^-)$
14:         Calculate TD loss $L_{\mathrm{TD}} = (Q_{\mathrm{tot}}^{\mathcal{P},\mathrm{QTRAN}}(\mathbf{h}, \mathbf{a}; \theta) - y^{\mathrm{target}})^2$
15:         Calculate $L_{\mathrm{opt}}$ using Eq. (12)
16:         Calculate $L_{\mathrm{nopt}}$ using Eq. (13)
17:         Update $\theta$ by minimizing $L = L_{\mathrm{TD}} + L_{\mathrm{opt}} + L_{\mathrm{nopt}}$
18:         Update $\theta^- = \theta$ with frequency $f$
19:     **end for**
20: **end for**

---

---

**Algorithm 5** Robust VDN with TV uncertainty set

---

1: Input robustness parameter $\rho$, target network update frequency $f$ and $\varepsilon$
2: Initialize replay buffer $\mathcal{D}$
3: Initialize $[Q_i^{\mathrm{rob}}]_{i \in [N]}$ with random parameters $\theta$
4: Initialize dual network $\eta_\xi$ with random parameters $\xi$
5: Initialize target parameters $\theta^- = \theta$
6: **for** episode $e = 1, \ldots, E$ **do**
7:     Observe initial state $s^0$ and observation $o_i^0 = \sigma_i(s^0)$ for each agent $i$.
8:     **for** $t = 1, \ldots, T$ **do**
9:         Each agent $i$ choose its action $a_i^t$ using $\varepsilon$-greedy policy.
10:         Take joint action $\mathbf{a}^t$, observe the next state $s^{t+1}$, reward $r^t$ and observation $o_i^{t+1} = \sigma_i(s^{t+1})$
            for each agent $i$
11:         Store transition $(\mathbf{h}^t, \mathbf{a}^t, s^t, r^t, \mathbf{h}^{t+1})$ in replay buffer $\mathcal{D}$
12:         Sample a mini-batch of transitions $(\mathbf{h}, \mathbf{a}, r, s, \mathbf{h}')$
13:         Calculate dual loss $L_{\mathrm{dual}}$ using Eq. (17)
14:         Update $\xi$ by minimizing $L_{\mathrm{dual}}$
15:         Sample another mini-batch of transitions $(\mathbf{h}, \mathbf{a}, s, r, \mathbf{h}')$ from $\mathcal{D}$
16:         Set $\bar{\mathbf{a}}' = [\arg\max_{a_i'} Q_i^{\mathrm{rob}}(h_i', a_i'; \theta^-)]_{i \in [N]}$
17:         Set $Q_{\mathrm{tot}}^{\mathcal{P},\mathrm{VDN}}(\mathbf{h}, \mathbf{a}; \theta) = \sum_{i \in [N]} Q_i^{\mathrm{rob}}(h_i, a_i; \theta)$
18:         Set $Q_{\mathrm{tot}}^{\mathcal{P},\mathrm{VDN}}(\mathbf{h}', \bar{\mathbf{a}}'; \theta^-) = \sum_{i \in [N]} Q_i^{\mathrm{rob}}(h_i', \bar{a}_i'; \theta^-)$
19:         Set $y^{\mathrm{target}} = r + \gamma(1 - \rho)\eta_\xi(s, \mathbf{a}) - \gamma[\eta_\xi(s, \mathbf{a}) - Q_{\mathrm{tot}}^{\mathcal{P},\mathrm{VDN}}(\mathbf{h}', \bar{\mathbf{a}}'; \theta^-)]_+$
20:         Calculate TD loss $L_{\mathrm{TD}} = (Q_{\mathrm{tot}}^{\mathcal{P},\mathrm{VDN}}(\mathbf{h}, \mathbf{a}; \theta) - y^{\mathrm{target}})^2$
21:         Update $\theta$ by minimizing $L_{\mathrm{TD}}$
22:         Update $\theta^- = \theta$ with frequency $f$
23:     **end for**
24: **end for**

---

**Algorithm 6** Robust QMIX with TV uncertainty set

---

1: Input robustness parameter $\rho$, target network update frequency $f$ and $\varepsilon$
2: Initialize replay buffer $\mathcal{D}$
3: Initialize $[Q_i^{\mathrm{rob}}]_{i \in [N]}$ and mixing network $f_\theta$ with random parameters $\theta$
4: Initialize dual network $\eta_\xi$ with random parameters $\xi$
5: Initialize target parameters $\theta^- = \theta$
6: **for** episode $e = 1, \ldots, E$ **do**
7:     Observe initial state $s^0$ and observation $o_i^0 = \sigma_i(s^0)$ for each agent $i$.
8:     **for** $t = 1, \ldots, T$ **do**
9:         Each agent $i$ choose its action $a_i^t$ using $\varepsilon$-greedy policy.
10:         Take joint action $\mathbf{a}^t$, observe the next state $s^{t+1}$, reward $r^t$ and observation $o_i^{t+1} = \sigma_i(s^{t+1})$
            for each agent $i$
11:         Store transition $(\mathbf{h}^t, \mathbf{a}^t, s^t, r^t, \mathbf{h}^{t+1})$ in replay buffer $\mathcal{D}$
12:         Sample a mini-batch of transitions $(\mathbf{h}, \mathbf{a}, r, \mathbf{h}')$
13:         Calculate dual loss $L_{\mathrm{dual}}$ using Eq. (17)
14:         Update $\xi$ by minimizing $L_{\mathrm{dual}}$
15:         Sample another mini-batch of transitions $(\mathbf{h}, \mathbf{a}, s, r, \mathbf{h}')$ from $\mathcal{D}$
16:         Set $\bar{\mathbf{a}}' = [\arg\max_{a_i'} Q_i^{\mathrm{rob}}(h_i', a_i'; \theta^-)]_{i \in [N]}$
17:         Set $Q_{\mathrm{tot}}^{\mathcal{P},\mathrm{QMIX}}(\mathbf{h}, \mathbf{a}; \theta) = f_\theta((Q_i^{\mathrm{rob}}(h_i, a_i; \theta))_{i \in [N]}, s)$
18:         Set $Q_{\mathrm{tot}}^{\mathcal{P},\mathrm{QMIX}}(\mathbf{h}', \bar{\mathbf{a}}'; \theta^-) = f_{\theta^-}((Q_i^{\mathrm{rob}}(h_i', \bar{a}_i'; \theta^-))_{i \in [N]}, s')$
19:         Set $y^{\mathrm{target}} = r + \gamma(1 - \rho)\eta_\xi(s, \mathbf{a}) - \gamma[\eta_\xi(s, \mathbf{a}) - Q_{\mathrm{tot}}^{\mathcal{P},\mathrm{QMIX}}(\mathbf{h}', \bar{\mathbf{a}}'; \theta^-)]_+$
20:         Calculate TD loss $L_{\mathrm{TD}} = (Q_{\mathrm{tot}}^{\mathcal{P},\mathrm{QMIX}}(\mathbf{h}, \mathbf{a}; \theta) - y^{\mathrm{target}})^2$
21:         Update $\theta$ by minimizing $L_{\mathrm{TD}}$
22:         Update $\theta^- = \theta$ with frequency $f$
23:     **end for**
24: **end for**

---

---

**Algorithm 7** Robust QTRAN with TV uncertainty set

---

1: Input robustness parameter $\rho$, target network update frequency $f$ and $\varepsilon$
2: Initialize replay buffer $\mathcal{D}$
3: Initialize $[Q_i^{\text{rob}}]_{i\in[N]}$, $Q_{\text{tot}}^{\mathcal{P},\text{QTRAN}}$ and $V_{\text{tot}}^{\mathcal{P},\text{QTRAN}}$ with random parameters $\theta$
4: Initialize dual network $\eta_\xi$ with random parameters $\xi$
5: Initialize target parameters $\theta^- = \theta$
6: **for** episode $e = 1, \ldots, E$ **do**
7:     Observe initial state $s^0$ and observation $o_i^0 = \sigma_i(s^0)$ for each agent $i$.
8:     **for** $t = 1, \ldots, T$ **do**
9:         Each agent $i$ choose its action $a_i^t$ using $\varepsilon$-greedy policy.
10:         Take joint action $\mathbf{a}^t$, observe the next state $s^{t+1}$, reward $r^t$ and observation $o_i^{t+1} = \sigma_i(s^{t+1})$
          for each agent $i$
11:         Store transition $(\mathbf{h}^t, \mathbf{a}^t, s^t, r^t, \mathbf{h}^{t+1})$ in replay buffer $\mathcal{D}$
12:         Sample a mini-batch of transitions $(\mathbf{h}, \mathbf{a}, r, s, \mathbf{h}', s')$
13:         Calculate dual loss $L_{\text{dual}}$ using Eq. (17)
14:         Update $\xi$ by minimizing $L_{\text{dual}}$
15:         Sample another mini-batch of transitions $(\mathbf{h}, \mathbf{a}, s, r, \mathbf{h}')$ from $\mathcal{D}$
16:         Set $\bar{\mathbf{a}}' = [\arg\max_{a_i'} Q_i^{\text{rob}}(h_i', a_i'; \theta^-)]_{i\in[N]}$
17:         Set $y^{\text{target}} = r + \gamma(1-\rho)\eta_\xi(s, \mathbf{a}) - \gamma[\eta_\xi(s, \mathbf{a}) - Q_{\text{tot}}^{\mathcal{P},\text{QTRAN}}(\mathbf{h}', \bar{\mathbf{a}}'; \theta^-)]_+$
18:         Calculate TD loss $L_{\text{TD}} = (Q_{\text{tot}}^{\mathcal{P},\text{QTRAN}}(\mathbf{h}, \mathbf{a}; \theta) - y^{\text{target}})^2$
19:         Calculate $L_{\text{opt}}$ using Eq. (12)
20:         Calculate $L_{\text{nopt}}$ using Eq. (13)
21:         Update $\theta$ by minimizing $L = L_{\text{TD}} + L_{\text{opt}} + L_{\text{nopt}}$
22:         Update $\theta^- = \theta$ with frequency $f$
23:     **end for**
24: **end for**

---

**Episode.** In `BuildingEnv`, each episode runs for 1 day, with 5-minute time intervals. That is, the horizon length $H = 288$, and the time interval length $\tau = 5/60$ hours. We set the discount factor $\gamma \simeq 0.997$ by using $H$ as the effective horizon length $H = \frac{1}{1-\gamma}$.

**State Space.** For a building with $N$ indoor zones, the state contains observable properties of the building environment at timestep $t$:

$$s(t) = (T_1(t), \ldots, T_N(t), T_E(t), T_G(t), Q^{GHI}(t), \bar{Q}^p(t)), \tag{30}$$

where $T_i(t)$ denotes zone $i$'s temperature at time step $t$, $\bar{Q}^p(t)$ is the heat acquisition from occupants' activities, $Q^{GHI}(t)$ is the heat gain from the solar irradiance, and $T_G(t)$ and $T_E(t)$ denote the ground and outdoor environment temperature.

**Observation Space.** For agent $i$, given the current state $s(t)$, its observation $o_i(t)$ is given by:

$$o_i(t) = \sigma_i(s(t))$$
$$= (T_i(t), T_E(t), T_G(t), Q^{GHI}(t), \bar{Q}^p(t)). \tag{31}$$

That is, agent $i$ can observe the tempature at zone $i$, the heat acquisition from occupants' activities, the heat gain from the solar irradiance, and the ground and outdoor environment temperature.

**Action Space.** At time $t$, agent $i$'s action is a scalar $a_i(t) \in [-1, 1]$, which sets the controlled heating supplied to zone $i$. The joint action $\mathbf{a}(t) = (a_1(t), \ldots, a_N(t))$

**Reward Function.** The objective is to reduce energy consumption while keeping the temperature within a given comfort range. Therefore, the reward function is a weighted average of these two goals:

$$r(t) = -(1-\beta)||\mathbf{a}(t)||_2 - \beta||T^{\text{target}} - T(t)||_2, \tag{32}$$

where $T^{\text{target}} = (T_1^{\text{target}}(t), \ldots, T_N^{\text{target}})$ are the target temperatures and $T(t) = (T_1(t), \ldots, T_N(t))$ are the actual zonal temperature, and $||\cdot||_2$ denote the $\ell_2$ norm. We use the same default hyperparameter $\beta$ across all experiments.

**Environmental Uncertainty.**   The environmental model and reward functions can differ from three climate types and locations (San Diego, Tucson, New York), which jointly decide the climate. Besides, `BuildingEnv` contains distribution shifts in the ambient outdoor temperature profile $T_E$ incurred by seasonal shifts.

## F.2   EXPERIMENTS SETUP.

We implement distributionally robust algorithms with two types of uncertainty sets ($\rho$-contamination Zhang et al. (2024), TV-uncertaintyPanaganti et al. (2022)) and three different value factorization methods (VDN (Sunehag et al., 2017), QMIX (Rashid et al., 2020), QTRAN (Son et al., 2019)). We also implement the GroupDR MARL algorithm from (Liu et al., 2025), also with these three different value factorization methods. Each experiment is run independently with 5 different random seeds.

**Hyperparameters.**   Training lasts 600 episodes with parameter updates every 2 steps and target updates every 25k steps. Replay buffer size is 10k; batch size is 64. We use $\varepsilon$-greedy exploration with $\varepsilon$ annealed from 1.0 to 0.01 over 120k steps. Hidden layer size is 64.

**Environment configurations.**   The environment configurations we use throughout the three experiments are shown in Table 3 where `Env_1` is the training environment, and the other environments are numbered in order of how much they differ from `Env_1`.

| Environment | Weather | Location |
|---|---|---|
| `Env_1` | Hot_Dry | Tucson |
| `Env_2` | Hot_Humid | Tampa |
| `Env_3` | Very_Hot_Humid | Honolulu |
| `Env_4` | Warm_Dry | El Paso |
| `Env_5` | Cool_Marine | Seattle |
| `Env_6` | Mixed_Humid | New York |

Table 3: Environment configurations for SustainGym `BuildingEnv`.

**Robust individual Q-networks.**   Each agent employs a recurrent Q-network (Hausknecht & Stone, 2015), encoding its local observation concatenated with the previous action (one-hot). Features pass through a fully connected layer with ReLU, followed by a single-layer LSTM (hidden size 64). The final hidden state is projected into Q-values. We optimize with RMSprop (lr = $5 \times 10^{-4}$).

**Mixing networks (QMIX).**   Following TorchRL (Bou et al., 2023), we use hypernetworks to generate state-dependent mixing weights while enforcing monotonicity. Per-agent Q-values are embedded, passed through ELU, and projected to a scalar joint Q. A state-conditioned bias is added via a two-layer MLP. Optimizer: RMSprop (lr = $5 \times 10^{-4}$).

**QTRAN Networks.**   We implement joint action-value and state-value networks as in Son et al. (2019). The joint action is encoded by concatenated one-hots, projected with ReLU, and optionally combined with agent features. The state-value network processes the global state and summed agent features in parallel, concatenates them, and outputs a scalar. Both use Adam (lr = $10^{-3}$).

**Dual networks (TV uncertainty).**   The global state and joint action (concatenated one-hots) are separately embedded via ReLU MLPs, concatenated, and passed through another ReLU layer before a final linear projection to a scalar. Optimizer: Adam (lr = $10^{-3}$).

**GroupDR training.**   GroupDR first fits a contextual-bandit-based worst-case reward estimator to estimate the worst-case reward, using data from `Env_1` to `Env_5` collected under a VDN-trained behavior policy. Using this estimator, we then train the individual Q-networks by randomly sampling episodes from `Env_1` to `Env_5`. (For fairness, `Env_6` is never used for training by any method.)

**Normalized Return.**   Because the reward in `BuildingEnv` is negative, we normalize the returns to $[0, 1]$ by the following transformation:

$$\text{Normalized\_return} = \frac{\text{Return} + 9000}{9000}. \tag{33}$$

**Experiment 1: climatic shifts.** During training, we train each algorithm in Env_1 by sampling episodes from Day 1 to Day 200. During evaluation, we evaluate each algorithm in Env_1 to Env_6, by sampling 50 episodes from Day 1 to Day 200 in each environment. This setup demonstrates distribution shift induced by climate differences.

**Experiment 2: seasonal shifts.** During training, we train each algorithm in Env_1 by sampling episodes from Day 1 to Day 200. During evaluation, we evaluate each algorithm in Env_1, by sampling 50 episodes from Day 400 to Day 600 in each environment. This setup demonstrates distribution shift induced by seasonality within the same location.

**Experiment 3: climatic and seasonal shifts.** During training, we train each algorithm in Env_1 by sampling episodes from Day 1 to Day 200. During evaluation, we evaluate each algorithm in Env_6, by sampling 50 episodes from Day 400 to Day 600 in each environment. This setup demonstrates the combined effects of climate and seasonal shifts.

## G DISCLOSURE OF LLM USAGE

To improve clarity and readability, we used a large language model (LLM) to assist in polishing the writing. The LLM was only employed for language refinement (e.g., grammar, style, and conciseness) and was not involved in designing methods, experiments, or drawing conclusions.

