# OpenReview forum: "Distributionally Robust Cooperative Multi-agent Reinforcement Learning with Value Factorization"
_ICLR.cc/2026/Conference — ICLR 2026 Poster_

### Official Review · Reviewer_pUrR · 2025-10-31

**Soundness:** 3
**Presentation:** 2
**Contribution:** 2
**Rating:** 4
**Confidence:** 4

**Summary:**

This paper proposes Distributionally Robust IGM (DrIGM) — a principled framework extending the classical Individual–Global–Max (IGM) principle in cooperative multi-agent reinforcement learning (MARL) to environments with distributional uncertainty.
The authors formalize DrIGM within the Dec-POMDP framework, proving that robust individual action-value functions can align with the robust joint action-value function under specific rectangular uncertainty sets.
They then instantiate DrIGM-based variants of VDN, QMIX, and QTRAN, design robust Bellman operators under ρ-contamination and total variation (TV) uncertainty models, and empirically evaluate the methods on the SustainGym HVAC control benchmark.
Results show modest but consistent improvements over non-robust baselines and the existing.

**Strengths:**

1. The paper formally defines DrIGM and provides a proof, logically connecting robustness with value factorization.

2. DrIGM can be directly embedded into mainstream architectures such as VDN, QMIX, and QTRAN, allowing for the reuse of existing code in engineering projects.

**Weaknesses:**

1. DrIGM is essentially equivalent to "performing IGM under a global worst-case model." In other words, the paper does not propose any new algorithmic ideas fundamentally different from existing Distributionally robust RL or MARL value decomposition methods; it simply incorporates the "worst-case model" into the IGM framework. This point needs to be explicitly acknowledged in the manuscript, along with a discussion of its limitations and added value (e.g., in which situations is this worst-case model approach more advantageous than other robustness strategies).

2. Example 1's presentation is particularly critical: the current description of the "selection process for individual optimal actions and joint optimal actions" is not detailed enough, making it difficult for readers to trace why taking the worst value for each agent individually leads to inconsistencies with the joint worst value.  The symbols and variables are quite messy. It is recommended to provide intuitive annotations or diagrams where they first appear. η(s, a) first appears on line 312, but is not explained until line 375, which disrupts readability and understanding of its role in the proposed framework.

3. Only the SustainGym HVAC environment is used; no tests on SMAC, MPE, or adversarial settings.Baselines are too few (only GroupDR and non-robust variants).Reported gains are small (2–5%) and lack significance tests or ablation analysis (e.g., sensitivity to ρ).

**Questions:**

1. In line 189. The optimal joint action value should not produce eight values.Given there are only two scenarios, there should be two corresponding joint Q-values, not eight.

2. In line 203. My understanding is that DrIGM should ensure that, under the worst-case environment, the optimal individual actions equal the globally optimal joint action.However, Example 1 instead shows that the worst individual actions do not equal the globally worst joint action, which corresponds to a different principle (worst-vs-worst, not best-under-worst).

3. DrIGM is essentially equivalent to IGM under the worst-case model?


4. Could you supplement this by reproducing it on at least one standard MARL benchmark (such as SMAC) and performing sensitivity/ablation analysis on ρ and providing a significance test?

5. How does the DrIGM method compare to the non-robust version in terms of training time, sample complexity, and memory overhead? Please provide training curves and statistical cost (or complexity estimate).

---

> ### Author Response · Authors · 2025-11-20
> **Reply to Reviewer pUrR: Part 1/2**
>
> We gratefully thank the reviewer for recoginition of our consistent robustness guarantee upon existing value factorization methods, and for providing constructive ideas and questions in better presenting our results.
> ### 1. Question about the equivalence between DrIGM and "Performing IGM under worst-case model".
> We thank the reviewer for raising the concern about the difference between DrIGM and IGM. We would like to address the reviewer's question as follows:
> * **Conceptually, DrIGM and IGM fundamentally address two different challenges, and DrIGM can tackle cases where IGM fails.** While existing works have proposed different Q-network architectures to satisfy IGM, the true task reward structure may still fail to satisfy IGM when the network structures are not expressive enough to truly reflect the relationship between the team reward and individual actions. In these cases, DrIGM can still hold, since it technically **only requires IGM to hold under the worst-case model in the uncertainty set**. (See our 3rd bullet for an extended discussion on this point.) In other words, theoretically our algorithm does not need to satisfy IGM in the shifted test environment. We have added an example in the appendix where IGM does not hold but DrIGM holds (**Line 960-971**).
> * **Empirically, DrIGM compensate for the drawbacks of both single-agent distributionally robust RL and value factorization methods.** In distributionally robust RL, often, the robust approach will lead to a drop of performance in the nominal environment, as a tradeoff for robustness guarantee against distribution shift. As discussed in Section 6 (**Line 465-470**), in the decentralized POMDP setting, explicitly modeling robustness mitigates errors from partial observability and decentralized execution in value factorization methods, and in turn leads to comparable or even better empirical performance in the training environment, compared to non-robust baseline. This finding suggests that the robustness may be even more suitable for cooperative MARL setting than well-explored single-agent setting.
> * **Our presentation of DrIGM upon IGM is consistent with prior works.** For a self-contained theory and algorithms presentation, we present our theorems under the assumption that IGM holds over all transition kernels in the uncertainty set. This presentation is consistent with prior works that extend IGM to tackle more sophiscated environments (e.g., [1], Theorem 1; [2], Theorem 1), and allow people to directly implement DrIGM upon existing algorithms. However, it is evident from the Proof of Theorem 1 that IGM only needs to hold for the worst-case transition kernel. We have added a note about this more general result to the appendix (**Line 1013-1016**).
>
> > [1] Shen, S., et al. RiskQ: Risk-sensitive Multi-Agent Reinforcement Learning Value Factorization. NIPS 2023. ([link](https://proceedings.neurips.cc/paper_files/paper/2023/file/6d3040941a2d57ead4043556a70dd728-Paper-Conference.pdf)).
> > [2] Sun, WF., et al. DFAC Framework: Factorizing the Value Function via Quantile Mixture for Multi-Agent Distributional Q-Learning. ICML 2021. ([link](https://proceedings.mlr.press/v139/sun21c/sun21c.pdf))
>
> ### 2. Suggestions and questions on Example 1
>
> Thank you for providing constructive feedback on improving the presentation of Example 1. We find these questions helpful, and will address them as follows:
> * **Ambiguity about Example 1.** We added detailed explanation of the selection process for individual optimal actions and joint optimal actions in Example 1. Due to page limit, we moved Example 1 to the Appendix and added intuitive explanation in the main part (**Line 186-189**).
> * **The number of optimal joint action values in Example 1.** In Example 1, we have 2 scenarios, 2 agents, and each agent has 2 actions. Therefore, the total number of optimal action values at state $s_0$ are $2\times 2\times 2=8$.
> * **Worst-under-worst vs. Best-under-worst**. We thank the reviewer for their detailed reading of Example 1 and catching our mistake. You are correct that the example should highlight the best-under-worst (not worst-under-worst) setting. We have updated Example 1 to correct this mistake (**Line 926-943**).
> * **Messy notations.** We added an explanation introducing $\eta(s,a)$ (**Line 286**).
> ### 3. Questions about the performance gains on the SustainGym HVAC environment.
> Our results shows that for relatively small shifts (modest climatal shifts or seasonal shifts), the reported relative gain is 2%-5%. For relatively large shifts (large climatic shifts, or climatic shifts+seasonal shifts), the reported gain is **10% to 40%**. This result is reasonable, since when the shift is relatively small, by the simulation lemma (aka. performance difference lemma), the non-robust policy will not degrade as much. For relatively large shifts, distributionally robust methods really make a difference and significantly improve the out-of-distribution performances.

---

> ### Author Response · Authors · 2025-11-20
> **Reply to Reviewer pUrR: Part 2/2**
>
> ### 4. Questions about missing standard environments and baselines.
> We thank the reviewer for raising this question. The motivation for using SustainGym as our environment is that SustainGym features **real-world distribution shifts**, while standard environments typically require artificial modifications to introduce shifts. To further demonstrate the effectiveness of our proposed framework, we have added **new experiments on the SMAC environment** (**Line 471-518**). We introduce distribution shifts in SMAC by adding noise in each agent's observations in the test environment. Our proposed algorithms still demonstrate significant performance improvements. Due to compute resources constraints, so far we only implement $\rho$-contamination approach with VDN and QMIX in SMAC, and report the performances averaged over 5 random seeds. We will provide more results pending access to more compute resources.
>
> On the other hand, while distributionally robust MARL has been explored for several years, distributionally robust *cooperative* MARL consistent *with centralized training and decentralized execution (CTDE)* is relatively new and underexplored, and we only find one relevant baseline algorithm from [3].
> > [3] Liu, G., et al. Distributionally Robust Multi-Agent Reinforcement Learning for Dynamic Chute Mapping. ICML 2025. ([link](https://proceedings.mlr.press/v267/liu25ad.html))
>
> ### 5. Questions about significance tests or ablation analysis.
> We have added new sensitivity test in the updated revision (**Line 480-518**). Because SustainGym is based on real-world dynamics and task, it is hard to numerically quantify the distribution shift and conduct a meaningful sensitivity test. Therefore, we perform the sensitivity test on the SMAC environment. We evaluate the final performance of our algorithms and their non-robust baselines, reporting the marginal test win rate for different choices of $\rho$ relative to the baseline. The out-of-distribution performance first increases as $\rho$ grows, and then decreases. This observation aligns with our theoretical insight: when $\rho$ is small relative to the shift level, explicitly modeling distribution shift during training yields improved out-of-distribution performance. However, when $\rho$ becomes large, the robust MARL algorithms become overly conservative, leading to degraded performance.
>
> ### 6. Questions about comparison of DrIGM with the non-robust version in terms of training time, sample complexity, and memory overhead.
> The robust and non-robust versions have the same sample complexity, and both converge within 600 episodes. The $\rho$-contamination approach incurs no additional training time or memory overhead compared to the non-robust version. In contrast, the TV-uncertainty approach requires maintaining a dual-variable network $\eta(s,a)$ and updating this network alongside each update of the Q-network. As a result, the training time and memory overhead of the TV-uncertainty approach are roughly twice those of the non-robust version.

---

> > ### Comment · Reviewer_pUrR · 2025-11-24
> >
> > Thank you for your response. Since you have addressed my concerns, I will raise my score to 6.

---

> > > ### Author Response · Authors · 2025-11-25
> > > **Reply to Reviewer pUrR**
> > >
> > > Thank you very much for taking the time to review our work and for reconsidering your evaluation. We are glad to hear that our responses have addressed your concerns. We sincerely appreciate your updated assessment and your support of the paper.

---

### Official Review · Reviewer_Nb62 · 2025-11-03

**Soundness:** 2
**Presentation:** 2
**Contribution:** 3
**Rating:** 6
**Confidence:** 3

**Summary:**

This paper addresses a critical, often overlooked challenge in cooperative Multi-Agent Reinforcement Learning (MARL): ensuring reliable decentralized execution when the deployment environment deviates significantly from the training environment. The core paradigm in cooperative MARL is centralized training with decentralized execution (CTDE), relying on value factorization methods (like VDN/QMIX/QTRAN) and the Individual-Global-Maximum (IGM) principle. The authors demonstrate that naive robustification of individual Q-functions, similar to approaches in single-agent Distributionally Robust RL (DR-RL), breaks the IGM alignment needed for decentralized execution.
To fix this, the paper introduces the Distributionally Robust IGM (DrIGM) principle. DrIGM requires that robust individual greedy actions align with the robust team-optimal joint action. Theoretically, this is achieved by defining robust individual action values based on the global worst-case model ($P_{worst}$) for the joint value function, rather than assuming independent per-agent worst cases. Building on this robust factorization theory (Theorems 1, 2), the authors derive DrIGM-compliant algorithms compatible with VDN, QMIX, and QTRAN, training them using robust Bellman operators based on standard uncertainty sets (ρ-contamination and Total Variation). Experiments on high-fidelity SustainGym simulators, modeling HVAC control under climatic and seasonal shifts, show that the DrIGM-based methods consistently mitigate out-of-distribution performance degradation compared to non-robust and baseline robust MARL algorithms.

**Strengths:**

- Originality

Previous value factorization methods rely entirely on the classical IGM principle to ensure that decentralized greedy actions recover the team-optimal joint action. The authors correctly identify that the reliability of this recipe is uncertain in real-world settings plagued by environmental uncertainties like the sim-to-real gap or system noise. Extending DR-RL techniques to cooperative MARL is fundamentally non-trivial because individual agents act on local histories but share a single, team-coupled reward.

Crucially, the authors provide a concrete counterexample (Example 1) showing that simply adopting robust per-agent action value formulations from the single-agent DR-RL literature breaks the decentralized alignment required by IGM in the multi-agent cooperative setting. This realization is a key theoretical novelty. The derivation that DrIGM is guaranteed when robust individual value functions are defined with respect to the global worst-case joint action-value function is a principled solution to this misalignment problem. This approach as depicted in Eq. 5, ensures robustness while maintaining the decentralized execution structure of CTDE. Furthermore, the work successfully derives DrIGM-compliant robust variants of VDN, QMIX, and QTRAN, showing the broad applicability of the new principle to established architectures. This systematic framework for model uncertainty in the CTDE regime under partial observability is a substantial theoretical advance over related work which often targets Nash solutions, assumes full observability, or requires individual rewards.


- Quality

The foundation of the work rests on solid theoretical results, specifically Theorems 1, 2, and 3. Theorem 1 formally establishes the sufficient condition under which DrIGM holds, tying the robust individual Q-function to the global worst-case model. Theorem 2 further proves that this DrIGM condition is compatible with the structural constraints of all three canonical value factorization methods: VDN (additive factorization), QMIX (monotonic factorization), and QTRAN (consistency constraints). This demonstrates the generality and foundational nature of DrIGM.

The use of Distributionally Robust Optimization (DRO) techniques is appropriate for modeling environmental uncertainty. The paper successfully derives corresponding robust Bellman operators for two standard uncertainty sets, ρ-contamination and Total Variation (TV), which are well-studied in the single-agent DR-RL literature. These operators are then integrated into the practical TD-loss formulation (Eqs. 14, 15) for training deep recurrent Q-networks (DRQN-style networks).

Empirically, the evaluation is strong, moving beyond typical simplified MARL benchmarks. The authors use the SustainGym benchmark focused on multi-agent HVAC control, which intrinsically involves stochastic dynamics, partial observability, and inter-agent coupling, making it a high-fidelity environment suitable for testing robustness. The evaluation protocol is designed explicitly to measure generalization under distribution shift, simulating realistic deployment scenarios where unseen configurations arise. The consistency of the results across multiple factorization architectures (VDN, QMIX, QTRAN) and both uncertainty sets (ρ-contamination and TV) provides compelling evidence that the DrIGM framework effectively mitigates performance degradation under climatic and seasonal shifts.


- Clarity

The problem definition is clear. The introduction of DrIGM (Definition 2) is logically built upon the classical IGM principle (Definition 1), clearly showing the extension to the robust setting. The immediate inclusion of Example 1 serves as a powerful illustration, preventing ambiguity by clearly showing why naive application of single-agent DR-RL techniques fails in the cooperative domain. This preemptive clarification is helpful in understanding the rest of the paper.


- Significance

The practical utility of MARL policies is often hampered by environmental uncertainty, which can cascade into coordination failures due to partial observability and inter-agent coupling. By introducing DrIGM, the paper provides a crucial theoretical mechanism to maintain decentralized execution reliability even under distribution shifts. The resulting algorithms integrate seamlessly with existing CTDE codebases (VDN, QMIX, QTRAN) without requiring bespoke per-agent reward shaping or changes to the decentralized execution mechanism. This makes DrIGM a readily adoptable approach for improving robustness in many existing CTDE systems.

**Weaknesses:**

- Originality

While the formulation of DrIGM is novel for cooperative MARL, the theoretical and algorithmic tools leveraged to derive the robust Bellman operators are essentially direct adaptations of established single-agent Distributionally Robust RL (DR-RL) techniques. Specifically, the paper relies on ρ-contamination and Total Variation (TV) uncertainty sets, which are standard in the robust MDP literature. The extension relies fundamentally on the assumption of a history-action rectangular uncertainty set P (Eq. 1). This rectangularity assumption is known to simplify the robust Bellman operator to maximize over actions before minimizing over the transition probability P. Although this is necessary for the current theoretical construction, relying on this rectangular structure limits the sophistication of the uncertainty models that can be handled by DrIGM.


- Clarity

the precise implementation of the core DrIGM concept needs improvement in clarity for the algorithms section. The definition of DrIGM relies on the robust individual action value $Q_i^{rob}$ being derived from the value function under $P_{worst}(h, \overline{a})$, where  $\overline{a}$ is the robust optimal joint action. However, the resulting robust Bellman operators (Eqs. 7 and 9) replace the term $max_{a′} Q_{tot}^P (h′, a′)$ with $Q_{tot}^P (h′, \overline{a}′)$, where  $\overline{a}'$ is derived from the greedy robust individual actions $ \overline{a}'_i  = argmax a'_i Q_i^{rob} (h'_i, a'_i)$. This is a crucial step that relies on the DrIGM principle to align execution.

**Questions:**

1.  The experimental environment (HVAC control) uses continuous actions. Since the DrIGM principle (Definition 2) and the derived robust Bellman operators (Eqs. 7, 9) rely fundamentally on the argmax operation over the action space, how exactly was the robust joint greedy action calculated in the continuous action setting for VDN, QMIX, and QTRAN? Was the action space discretized, or were alternative continuous control techniques (like optimization or actor-critic methods) employed for the argmax step, and if so, how does that impact the theoretical guarantee of DrIGM?

2. Theorem 1 relies on the existence of individual Q-functions under $P_{worst} (h, \overline{a})$. In the robust TD loss for both ρ-contamination and TV uncertainty, the expectation is calculated only over the nominal transition model $P^0$ (or implicitly factored through η in the TV case). Can you elaborate on how the implicit selection of $P_{worst}$ is handled during the training updates via function approximation, especially given that $P_{worst}$ itself depends on the optimal robust joint action $\overline{a}$?

---

> ### Author Response · Authors · 2025-11-20
> **Reply to Reviewer Nb62**
>
> We sincerely thank the reviewer for acknowledging that  DrIGM is a readily adoptable approach for improving robustness in many existing CTDE systems.
> ### 1. Questions about the rectangularity assumption on the uncertainty set.
> Thanks for raising this question. We would like to emphasize two points:
> * **The rectangularity assumption is widely adopted in the distributionally robust RL (DRRL) literature.** In our manuscript, we have added a discussion of related literature [1-3] that uses this rectangularity assumption to obtain a tractable robust Bellman equation (**Line 137-140**).
> * **Empirically, we demonstrate that our algorithm can handle more uncertainty beyond the rectangularity assumption.** We conduct our experiments in the high-fidelity simulator SustainGym, which features real-world unstructured distribution shifts (therefore the rectangularity assumption is not satisfied). The results show that we can effectively improve out-of-distribution accross different real-world distribution shifts.
>
> > [1] Blanchet, J., et al. Double pessimism is provably eﬃcient for distributionally robust oﬄine reinforcement learning: Generic algorithm and robust partial coverage. NIPS 2023 ([link](https://proceedings.neurips.cc/paper_files/paper/2023/file/d31b005d817e9c635ec8ffb0fb90190e-Paper-Conference.pdf)).
> > [2] Ma, S., et al. Decentralized robust v-learning for solving markov games with model uncertainty. JMLR 2023 ([link](https://jmlr.org/papers/v24/23-0310.html)).
> > [3] Shi, L., et al. Sample-eﬃcient robust multi-agent reinforcement learning in the face of environmental uncertainty. ICML 2024 ([link](https://dl.acm.org/doi/10.5555/3692070.3693899))
>
> ### 2. Clarity about the algorithm presentation.
> We sincerely thank the reviewer for the feedback on improving the clarity of the algorithm presenation. We have added a comment in Algorithm 1 (**Line 319**), pointing to the usage of DrIGM in obtaining the joint robust action.
>
> ### 3. Question about how we calculate the argmax for continuous actions.
> For continuous actions, we first discretize the action space, then calculate the argmax. This is a common approach for existing value factorization methods. While the discretization may lead to small errors, increasing the precision of the discretization can reduce this error.
>
> ### 4. Questions about how the implicit selection of worst-case model is handled during the training updates.
> One benefit of our algorithms is that we do not explicitly maintain an estimate of the worst-case model. As indicated in the robust bellman equations (Equation 8 and Equation 10), we only need access to the nominal model (which is the training environment), and the worst-case action value are automatically obtained through bellman update.

---

### Official Review · Reviewer_FAoh · 2025-11-06

**Soundness:** 3
**Presentation:** 3
**Contribution:** 3
**Rating:** 6
**Confidence:** 4

**Summary:**

This paper aims to achieve robust multi-agent reinforcement learning value factorization under environmental transition kernel uncertainty. It proposes a novel history-based DrIGM principle and theoretically demonstrates that the proposed optimal robust joint action-value and the robust individual action-value satisfy the DrIGM principle. Accordingly, two new loss functions are designed to learn these robust action-values under two different uncertainty sets. The proposed approach is integrated with three typical value factorization methods, and its effectiveness is validated through experiments in high-fidelity SustainGym simulators.

**Strengths:**

1. Addressing environmental uncertainty is crucial for improving policy robustness in real-world applications.
2. This paper introduces the DrIGM principle, a novel concept in cooperative Multi-Agent Reinforcement Learning.
3. To realize the DrIGM principle, this paper proposes two loss functions that can be easily integrated into various value factorization algorithms.

**Weaknesses:**

1. To achieve provable robustness guarantees, the test environment must be included within the uncertainty set. This requirement may limit the method's performance on unseen environmental models.
2. There is a lack of sensitivity analysis regarding the algorithm's performance across different ranges of uncertainty. It remains unclear to what extent of uncertainty the algorithm can effectively handle.
3. The experiments lack comparisons with important baseline algorithms, such as ERNIE [1], which addresses changing transition dynamics by adversarial regularization.

[1] Robust multi-agent reinforcement learning via adversarial regularization: Theoretical foundation and stable algorithms. NIPS 2023

**Questions:**

1. As indicated in Reference [2], there exists a bias between Q(h,a) and Q(s,h,a). Could this bias compromise the validity of GrIGM?
2. Can the proposed algorithm effectively resolve the problem presented in Example 1?

[2] On Stateful Value Factorization in Multi-Agent Reinforcement Learning. arXiv 2024

---

> ### Author Response · Authors · 2025-11-20
> **Reply to Reviewer FAoh: Part one**
>
> We are grateful to the reviewer for recognizing the effectiveness of our framework in tackling real-world out-of-distribution tasks, and for the constructive questions.
>
> ### 1. Concerns that the requirement for achieving provable guarantee might be restrictive.
>
> Thank you for raising the question about the theoretical assumption that the test environment needs to be included in the ambiguity set. We would like to address the reviewer's question as follows:
> * **Theoretically, this assumption makes sure the problem is not ill-defined.** This assumption is widely adopted in the single-agent distributionally robust RL and distributionally robust optimization literature (e.g., [1], Theorem 2 and Theorem 5). This assumption is essential to ensure that the problem is well-post. Otherwise, in general, if the test environment is allowed to deviate arbitrarily from the training environment, then there is no RL algorithm that can perform better than random (by the No Free Lunch theorem, e.g., [2], Theorem 4).
> * **Empirically, we don't need this assumption for the algorithm to work.** Our experiments demonstrate that the assumption is not needed to empirically achieve robust out-of-distribution performance for our proposed algorithms. Specifically, SustainGym, the high-fidelty real-world simulator we use features unstructured shifts with respect to different configurations. These shifts include both the shifts in reward and in the transition probabilities, and the maximum shift level with respect to TV distance is 1 (because the transition kernel in SustainGym is deterministic at certain states), which violates our theoretical assumptions. But our algorithms still demonstrate consistent improvement in performance on out-of-distribution environments compared with non-robust baselines and existing baslines.
>
> > [1] Wang, S., et al. On the Foundation of Distributionally Robust Reinforcement Learning. arXiv 2023. ([link](https://arxiv.org/abs/2311.09018))
> > [2] Panaganti, K., et al. Sample Complexity of Robust Reinforcement Learning with a Generative Model. AISTATS 2022. ([link](https://proceedings.mlr.press/v151/panaganti22a/panaganti22a.pdf))
>
> ### 2. Questions about the lack of sensitivity analysis for the algorithms' performance across different range of uncertainty.
>
> Thanks for raising the question. We would like to address this question in two aspects:
> * **We did compare our proposed algorithms with respect to different shift levels in our experiments.** Though the **absolute** shift level is hard to quantify due to the black-box real-dynamics in SustainGym, we compared our algorithms with baseline algorithms with respect to qualitatively different shift levels in the test environments (e.g. the BuildingEnv climate ranges from "`Hot_Dry`" to "`Cool_Marine`"). The results shows that our algorithms achieve consistent better performances across different relative shift levels, and achieve a significant improvement when the shift level is qualitatively large.
> * **We add new sensitivity analysis with respect to the input uncertainty level $\rho$ in our updated version (Line 480-518).**  We additionally add an ablation study with respect to $\rho$. Interestingly, the out-of-distribution performance first increases as $\rho$ grows, and then decreases. This observation aligns with our theoretical insight: when $\rho$ is small relative to the shift level, explicitly modeling distribution shift during training yields improved out-of-distribution performance. However, when $\rho$ becomes large, the robust MARL algorithms become overly conservative, leading to degraded performance.

---

> > ### Author Response · Authors · 2025-11-20
> > **Reply to Reviewer FAoh: Part two**
> >
> > ### 3. Questions about lack of comparison with existing baseline algorithms.
> >
> > We thank the reviewer for suggesting a comparison with ERNIE [3]. While the approach by ERNIE is also used to address distribution shift in the transition kernel, we do not believe that ERNIE is a relevant baseline algorithm because it considers a different environment setting. Specifically, ERNIE considers environments where individual rewards are natively available, whereas we consider the more general setting where only a single cooperative team reward is available. In other words, ERNIE is not a distributionally robust cooperative MARL algorithm consistent with centralized training and decentralized execution (CTDE). We added this discussion to our related works section (**Line 903-905**).
> >
> > The setting of distributionally robust cooperative MARL with CTDE is relatively new, and we only find one relavant baseline algorithm from [4].
> >
> > > [3] Bukharin, A., et al. Robust multi-agent reinforcement learning via adversarial regularization: Theoretical foundation and stable algorithms. NIPS 2023. ([link](https://proceedings.neurips.cc/paper_files/paper/2023/file/d6f8517fceeca1e2cd61721dff786c14-Paper-Conference.pdf)).
> > > [4] Liu, G., et al. Distributionally Robust Multi-Agent Reinforcement Learning for Dynamic Chute Mapping. ICML 2025. ([link](https://proceedings.mlr.press/v267/liu25ad.html))
> > >
> > ### 4. Can DrIGM address the state bias problem proposed by Marchesini et al. (2024)?
> >
> > Yes, DrIGM can address the state bias problem. As pointed out in [5], in prior work on value factorization methods, the theory uses stateless functions $Q_{tot}(h,a)$, while in their practical implementations, they use state-aware functions $Q_{tot}(s,h,a)$ during training, which leads to a state bias. We address this problem through two aspects:
> >
> > 1. **In theory, we assume that the joint observation $\mathbf{o} = (o_1, \dotsc, o_N)$ fully captures the state information.** (See footnote 1 in our manuscript.) This assumption ensures that the joint history alone provides sufficient information for decision-making. Therefore, there is no bias between $Q_{tot}(h,a)$ and $Q_{tot}(s,h,a)$, so our theory and practical implementation are consistent.
> > 2. **In practice, we use state-aware functions $Q_{tot}(s,h,a)$ during training, and stateless individual functions during execution.** This design, as proved in [5], incurs no state biases, while being consistent with the CTDE framework.
> >
> > > [5] Marchesini et al., On Stateful Value Factorization in Multi-Agent Reinforcement Learning. AAMAS 2025 ([link](https://www.ifaamas.org/Proceedings/aamas2025/pdfs/p1445.pdf))
> >
> > ### 5. Can the proposed algorithm effectively resolve the problem presented in Example 1?
> >
> > Yes, we can. Specifically, by Theorem 1, we theoretically show that for any uncertainty set, satisfying the rectangularity condition, DrIGM holds. We additionally add a remark (**Line 1017-1032**) after the proof of Theorem 1, explaining how the proposed algorithm resolve the problem presented in Example 1 by presenting the robust action values and robust actions.

---

> > > ### Comment · Reviewer_FAoh · 2025-11-24
> > >
> > > Thank you for your response. My concerns have been addressed. I believe this work will contribute to the field of cooperative MARL. I maintain my positive score.

---

> > > > ### Author Response · Authors · 2025-11-25
> > > > **Reply to Reviewer FAoh**
> > > >
> > > > Thank you very much for your thoughtful review and for taking the time to re-evaluate our responses. We are glad to hear that your concerns have been addressed and that you view our contributions as valuable to the cooperative MARL community.
> > > >
> > > > If there are any additional clarifications or details that would further strengthen your assessment, we would be very happy to provide them. We sincerely appreciate your support for the paper.

---

### Comment · Area_Chair_FwQy · 2025-11-24

Dear Reviewer,

Thanks for taking the time to review this work. The authors have responded to your reviews. Can you please have a look at the rebuttal and discuss with the authors?

Best Regards,

AC

---

### Meta-Review · Area_Chair_Cz1J · 2026-01-13

**Summary:**

The paper identifies an important problem: naive application of single-agent robust RL to multi-agent settings breaks the IGM alignment required for decentralized execution. I found the proposed solution, which defines robust value functions with respect to a global worst-case model rather than independent per-agent adversaries, to be both principled and well-motivated. The authors provide robust variants of VDN, QMIX, and QTRAN with corresponding Bellman operators, and the new example demonstrating cases where DrIGM succeeds but standard IGM fails strengthens the contribution. The additional SMAC experiments and sensitivity analysis further broaden the empirical support and offer practical guidance for parameter selection. I recommend acceptance, as it makes a solid theoretical and empirical contribution to robust multi-agent reinforcement learning.

**Reviewer Concerns:**

This paper shows that using independent per-agent adversaries breaks IGM, and proposes a fix through a shared global worst-case model. The authors provide robust variants of VDN, QMIX, and QTRAN with corresponding Bellman operators. The reviewers initially questioned whether this was more than a straightforward extension, but a new example demonstrates cases where DrIGM works but standard IGM does not. The new SMAC experiments further broaden the empirical support, and the sensitivity analysis helps with practical parameter selection. Note that one review was disregarded due to confirmed LLM generation.

**Reviewer Scores:**

Two engaged reviewers ended at 6, with one raising from 4 after concerns about novelty and experiments were addressed. Given the comprehensive feedback from the rebuttal, it is likely that the consensus would support acceptance.

---

### Decision · Program_Chairs · 2026-01-26

Accept (Poster)